# Acid-tolerant injectable bioadhesive for sutureless repair of large gastric perforation

Ze Wang [1,2,3,6], Bo Cao[1,3,6], Longsong Li[4,6], Hao Cui[1], Bo Wei[1,3] ✉,
Jianxin Cui[1,3] ✉ & Xing Wang [2,5] ✉

Bioadhesives represent promising alternatives to sutures towards gastric perforation management, however, significant challenges persist concerning instant wet adhesion and durable stability in gastric perforation sealing, particularly in direct contacting with acidic gastric fluids on large perforation injuries. Here we report an injectable acid-tolerant hydrogel composed of FDA-approved components for sutureless repair of large gastric defects. The hydrogel displays rapid in situ gelation, instant wet adhesion, and high burst pressure for efficient sealing despite excessive mechanical challenges, tissue irregularities and gastric juices. The enhanced hydrogen bonding interactions among amide-linked skeleton enable robust acid-tolerant interfaces to accommodate durable adhesion under the fluidically, chemically and mechanically dynamic in-vivo environments. A larger-scale porcine gastric perforation is applied to validate the sealing efficacy via a combined laparoscopic-endoscopic technique. The negligible postoperative adhesion, suppressed inflammation and interference-free transcriptome and microbiome verify the therapeutic outcomes. The proposed bioadhesives hold great promise for clinical treatment of digestive diseases.

Gastric perforation is a potentially life-threatening surgical emergency in which patients suffer from severe medical complications like peritonitis, sepsis, abscess, septic shock, and even multiple organ dysfunction syndromes, resulting in more than 30% increase in mortality[1–3]. Surgical sutures and staplers are the predominant clinic operations by means of laparoscopy or laparotomy for the emergent intervention of gastric perforation, but they always require delicate control and long surgical time that is problematic in most emergency circumstances. Furthermore, their inherent disadvantages always bring serious adverse potentials, such as additional gastric trauma, pointwise closure, delayed wound healing, anastomotic imperfections, postoperative adhesion, long-term inflammatory response, and other complications[4–6]. Hence, surgical repair to provide mechanical sealing and favorable healing remains an ongoing challenge, thus

highlighting the critical importance and urgency on developing effective solutions to prevent fluid leaks and accelerate wound cares.

Bioadhesives have been promising as alternatives to sutures and staples for gastric perforation management because of their easy operation, good cytocompatibility, strong tissue adhesion, and adaptive flexibility even in the wet and dynamic biological environments[7–16]. Of these, cyanoacrylate and fibrin glue are the two most commonly surgery representatives for medical settings, but the potential tissue toxicity of brittle cyanoacrylate and weak adhesion of degradable fibrin glue compromise their treatment outcomes, let alone the harsh gastric sealing with a strong acid environment and continuously dynamic movements[17,18]. Although several developed hydrogel bioadhesives have been demonstrated for providing tough adhesion to wet tissues and promoting gastric tissue repair in recent years, critical

[1]Department of General Surgery, The First Medical Center, Chinese PLA General Hospital, Beijing, China. [2]Beijing National Laboratory for Molecular Sciences, Institute of Chemistry, Chinese Academy of Sciences, Beijing, China. [3]Medical School of Chinese PLA, Beijing, China. [4]Department of Gastroenterology, The First Medical Center, Chinese PLA General Hospital, Beijing, China. [5]University of Chinese Academy of Sciences, Beijing, China. [6]These authors contributed equally: Ze Wang, Bo Cao, Longsong Li. ✉e-mail: weibo@301hospital.com.cn; cuijx_doctor@163.com; wangxing@iccas.ac.cn

challenges remain for clinical application regarding the potential toxicity, undesirable swelling and subsequent gradual deterioration in strength and interfacial robustness[19–23], especially in the long-term contact with acidic fluids in vivo and the probable leakage of gastric contents and peritonitis from larger diameter of gastric perforations[24–27], which demands a combination of instant strong wet adhesion, acid/enzyme tolerance, low swelling, and long-term mechanical/adhesive stability under dynamic peristalsis. This is because achieving all these properties simultaneously remains difficult; many hydrogel systems are facing trade-off issues, such as maintaining long stability in harsh gastric environments and achieving controlled and harmless degradation[28–30]. The design of bioadhesives that seamlessly integrates long-term acid/enzyme resistance with predictable and benign degradation profiles into nontoxic byproducts is highly desirable for translational applications[31]. Besides, in situations when direct placement of hydrogels is inaccessible, in situ injectable hydrogel formation with ultrafast gelation and sufficient adhesion to the target sites is desirable, particularly in the irregular surface of gastric tracts and the fluidically, chemically and mechanically dynamic in vivo environment. More importantly, the gastric tissue, in fact, can quickly update the mucosa and accelerate the gastric healing with the assistance of hydrogel bioadhesives in cases of gastric perforations defects with relatively small size (ca. 1 cm)[32–36], whereas larger gastric perforations are routinely challenged and rarely reported so far. Consequently, the development of injectable bioadhesives with favorable biocompatibility, instant wet adhesion, low swelling, tissue-matching biodegradability and acid-tolerance capacity under complex chemical and mechanical environments is still highly anticipated.

Poly(ethylene glycol) (PEG), as a nontoxic and nonimmunogenic FDA-approved synthetic polymer, has been widely employed to construct various hydrogel adhesives in biomedical applications[37–40]. In our previous work, we reported an in situ injectable sutureless sealant (tetra-PEG) by an ammonolysis reaction between the tetra-armed poly(ethylene glycol) amine (tetra-PEG-NH$_2$) and tetra-armed poly(ethylene glycol) succinimidyl succinate (tetra-PEG-SS). This developed sealant showed ultrafast gelation, superior biocompatibility, high mechanical strength, strong tissue adherence and compliance to wet tissue for sutureless dural closure[41]. These fascinating features meet the basic properties required for medical sealants, but it is difficult to persistently adhere to the target sites under strong acid, digestive enzymes, and mechanically dynamic environments. Herein, in this study, we develop another kind of injectable PEG-based OSSA hydrogel bioadhesive based on the octa-armed poly(ethylene glycol) succinimidyl succinamide (octa-PEG-SSA) and octa-armed poly(ethylene glycol) amine (octa-PEG-NH$_2$) for the repair of large gastric perforation (3 cm). In addition to retaining innate features of injectable shape adaptability, ultrafast gelation, good biocompatibility and instant wet adhesion, the stable amide-linked skeleton and strong hydrogen bonding interactions within the dense crosslinking network furnish the OSSA hydrogel with gastric fluid resistance, low swelling ratio and slow degradable deterioration in strength and interfacial robustness, thus accommodating the long-term adhesion on the gastric tissues under wet and dynamic motion of the porcine stomach. We envision that the biodegradation rate of the OSSA hydrogel can be well compatible with the gastric tissue healing without causing obvious inflammatory responses. A combination of the transcriptome and microbiome has been used to evaluate the superiority, biosafety, and efficacy of hydrogel bioadhesive for sutureless repair of gastric defects in addition to the morphological and histological evaluation (Fig. 1). Collectively, this bioadhesive embodies multifunctionalities to not only synergistically solve the above-mentioned key limitations of sutures and commercially available tissue sealants (Fig. 2a, b), but also initiatively achieve instant wet adhesion and durable closure of large-size porcine gastric perforations via laparoscopic delivery and intuitively

monitor the long-term sealing efficacy in real time via the combined laparoscopic-endoscopic technique, consequently offering a clinically feasible and effective therapeutic approach for sutureless repair of large gastric perforation.

## Results

### Design strategy

In views of the intrinsic properties of ammonolysis reaction between the active ester and amine groups, the injectable OSSA hydrogel was quickly formed within 5 s through mixing the octa-PEG-SSA and octa-PEG-NH$_2$ solutions (Supplementary Figs. 1 and 2). During this gelling process, OSSA bioadhesive could quickly and robustly adhere to wet gastric tissues based on a dry cross-linking mechanism (Fig. 2c). In brief, the ultrafast sol–gel transition process allowed the hydrogel bioadhesives with rapid absorption and drying of interfacial water on wet gastric tissues upon contact. Subsequently, the residual succinimidyl-active ester groups facilitated rapid and robust adhesion to the gastric tissue surface based on covalent cross-linking via amide bonds, which could efficiently resist gastric motility and harsh juices for providing instant, fluid-tight, and sutureless sealing of large gastric perforation with straightforward application (Fig. 2d). In addition, the succinimidyl-active ester groups involved in the noncontact surface could be easily eliminated by a short-time water flushing, enabling a nonadhesive interface to the surrounding tissues due to the anti-protein adsorption function and anti-cell adhesion feature of PEG itself (Fig. 2e). This double-sided or Janus-like bioadhesive was significant important for surgeons in repair of the gastrointestinal tract and postoperative adhesion prevention.

### Physicochemical properties of the OSSA hydrogel

Taking consideration the gelation time, syringe ability, mechanical property, adhesive strength, and degradation period, we chose the octa-PEG hydrogel with a concentration of 15 wt% to satisfy the superior performance requirement (Fig. 3a, b). In order to facilitate the operation in rheological instrumentation, we used an 8% concentration instead to slow down the gelation time. Figure 3c showed an abrupt increase of both storage modulus $G'$ and viscosity $\eta$ (associated with dissipative modulus $G''$) only after 18 s, indicating the gelation of polymer solution and reflecting faster gelation time of octa-PEG hydrogel at 15 wt% concentration. During this gelling process, high adhesion strength onto the tissue was meanwhile achieved via the formation of chemical linkage between the tissue protein amines and succinimidyl-activated octa-PEG-SSA polymer. In addition, the rapid in situ gelation without the need of catalysts or irradiation and facile injection implementation with ready-to-use feature and robust tissue adhesion were significantly essential for the convenient and instant closure of the irregular gastric defects to prevent bleeding and leakage of bodily fluids. The flexible OSSA hydrogel was entirely transparent that played an important role in the unobstructed examination of the underlying tissues during surgery. Its well-defined architectural networks enabled the accessible nutrient exchange and cell support (Fig. 3d) and competitive mechanical strength. The high compressive strength and repeated compression behaviors at strains of 10, 20 and 30% verified its practicability and stability even when there were outside forces after application (Fig. 3e, f). In addition, the cyclic stress–strain compressive curves at a strain of 30, 50, and 70% (Fig. 3g and Supplementary Fig. 3) and cyclic stress–strain tensile curves at a strain of 40, 80, and 120% (Fig. 3h and Supplementary Fig. 4) further validated the good ductility and fatigue resistance of the OSSA hydrogel that could require the change in volume of the stomach before and after eating. The rheological result also proved that the OSSA hydrogel could maintain stable in a wide sweeping frequency, and the greater $G'$ than $G''$ over the tested frequency range further indicated its stability after anchor to the gastric tissues even under dynamic environments (Fig. 3i).

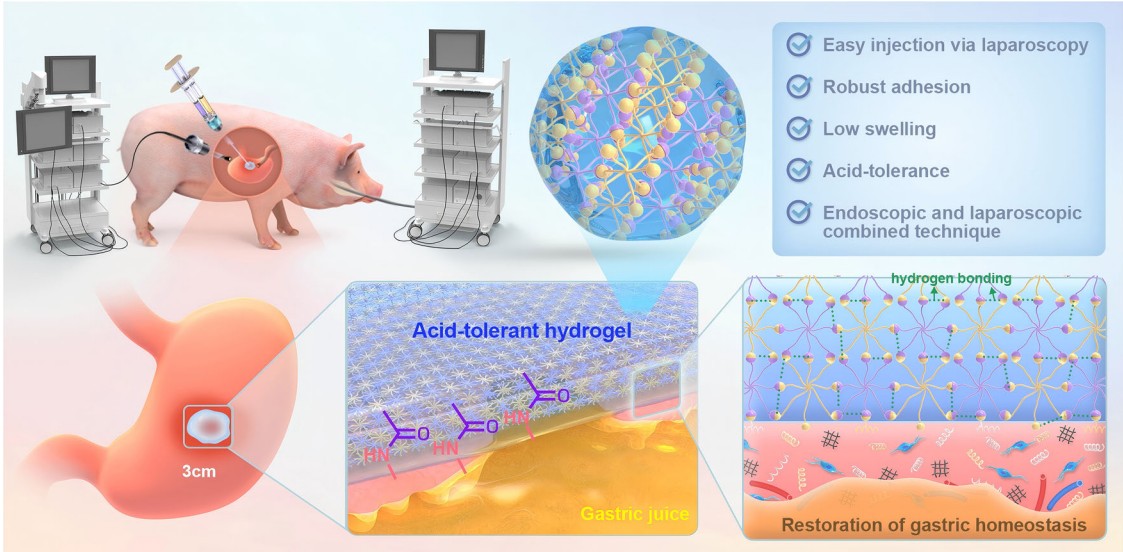

**Fig. 1 | Schematic diagram illustrating the injectable OSSA hydrogel bioadhesive for sutureless repair of large gastric perforation.** By means of the combined laparoscopic-endoscopic technique, this acid-tolerant OSSA hydrogel enables effective sealing against gastric juice and restoration of gastric homeostasis for sutureless repair of large gastric perforation.

To evaluate the adhesive performance of OSSA hydrogel, fresh porcine skin tissue was used as the biological substrate to mimic human tissues. Owing to the strong chemical bonding between the succinimidyl-active ester moieties on hydrogel and the amino groups of tissue proteins, the injectable OSSA hydrogel could firmly adhere to the wet porcine skin without detachment and maintain original morphology under different external stresses (stretching, bending, and twisting, Fig. 3j), indicating the formation of robust interface adhesion and sufficient deformability to tolerate the gastric motility. Wherein, the OSSA adhesion onto the porcine tissue cannot be washed away even by high-pressure flushing water (Fig. 3k and Supplementary Movie 1), which was of particular importance in clinical applications where multiple water irrigation during the laparoscopic or gastroscopic surgery may be required. To quantitatively evaluate the adhesive properties, the shear strength and interfacial toughness of the OSSA hydrogel to tissue was further quantitatively tested by lap shear and 180° peel tests (Fig. 3l, m). The shear adhesive strength of OSSA hydrogel could achieve $34.0 \pm 2.3\,kPa$, superior to the commercially available tissue adhesives and sealants, including the Coseal $(10.3 \pm 0.9\,kPa)$, fibrin glue $(10.4 \pm 0.6\,kPa)$, and cyanoacrylate glue (Histoacryl, $23.6 \pm 1.5\,kPa$). Similarly, the interfacial toughness was more than $118.2 \pm 10.2\,J\,m^{-2}$ in comparison with those of the Coseal $(23.1 \pm 3.4\,J\,m^{-2})$, fibrin glue $(24.8 \pm 2.9\,J\,m^{-2})$, and Histoacryl $(39.7 \pm 3.3\,J\,m^{-2})$ adhesives. Besides, the high bulk mechanical strength of OSSA hydrogel (1.1 MPa, Fig. 3e) also contributed to the satisfactory adhesive strength, because the total adhesive performance of tissue adhesive was generally governed by the interface adhesive and cohesive strengths (bulk mechanical property). A distinct and coherent adhesive-tissue interface was observed from the SEM images (main view, Fig. 3n), validating the formation of robust interfacial adhesion. In fact, in addition to the generation of amide linkages between the tissue and deposited bioadhesives, given that the OSSA sealant was applied by placing precursor solution onto the surface of porcine gastric tissues (top view, Fig. 3n), the free and flexible PEG chains could penetrate the tissues and form the physical entanglements with collagen fibers to synergistically improve the interface adhesion, thus collectively contributing to the robust interfacial adhesion.

It was mentioned that although many published literatures on hydrogel bioadhesives had demonstrated their tough adhesion to wet tissues and sealing effects on gastric defects, a key contradiction between the interfacial stability and hydrogel biodegradability had always been avoided or not discussed, but simply emphasized the adhesion strength and short-term tolerance of the gastric fluid environment using in vitro experiments. In fact, human gastric juice contained a lot of acidic fluids, digestive enzymes, intrinsic factors and mucoproteins that could rapidly destroy the majority of natural polysaccharides linked by glucoside bonds and synthetic degradable polymers linked by ester bonds through acidic or enzymatic hydrolysis, thus accelerating the degradability of these bioadhesives and impairing their sealing effects before the gastric healing (>2 weeks). While the other acid-tolerant synthetic hydrogels like poly(acrylic acid)-based adhesives that could maintain good structural stability in the harsh gastric fluid environment may be difficult to degrade in vivo, thereby hindering new tissue formation. So, we comprehensively estimated the structural stability and durable wet adhesion of OSSA hydrogel in human gastric juices with negligible change in mass, size and transparency and maintainable tissue adhesion within 14 days (Supplementary Figs. 5 and 6). These unique results indicated that an amide-based OSSA network was a fundamental redesign for constructing stable networks to directly conquer the clinical challenge of robust adhesive stability for a long period, particularly in direct contacting with acidic gastric fluids.

## Ex vivo gastric perforation sealing performance

Robust adhesion to wet and dynamic interfaces is essential for effective wound closure because of the long direct contact with harsh gastric juice and dynamic gastric movements. Burst pressure was critical parameter to evaluate the closure capacity of the OSSA adhesive against gastric fluid or air leakage due to the luminal pressure[25,29]. In views of the easy manipulations of colonic closure for quantitative assessment, the OSSA hydrogel was employed to seal a penetrating hole with a diameter of 4 mm on the wet porcine colon followed by pumping into PBS solution at a flow velocity of 2 mL/min (Fig. 4a). The burst pressure of OSSA adhesive reached ~19 kPa (142 mmHg), which far exceed the commercially available Coseal, fibrin glue, and Histoacryl (Supplementary Fig. 7), as well as the in vivo gastrointestinal pressure (0.5–4.0 kPa, 3.7–30 mmHg) of normal human[42]. Moreover, the OSSA hydrogel could maintain a firm sealing interface in the porcine colon to resist continuous motion unless it was subjected to violent puncture and peeling (Fig. 4b and Supplementary Movie 2),

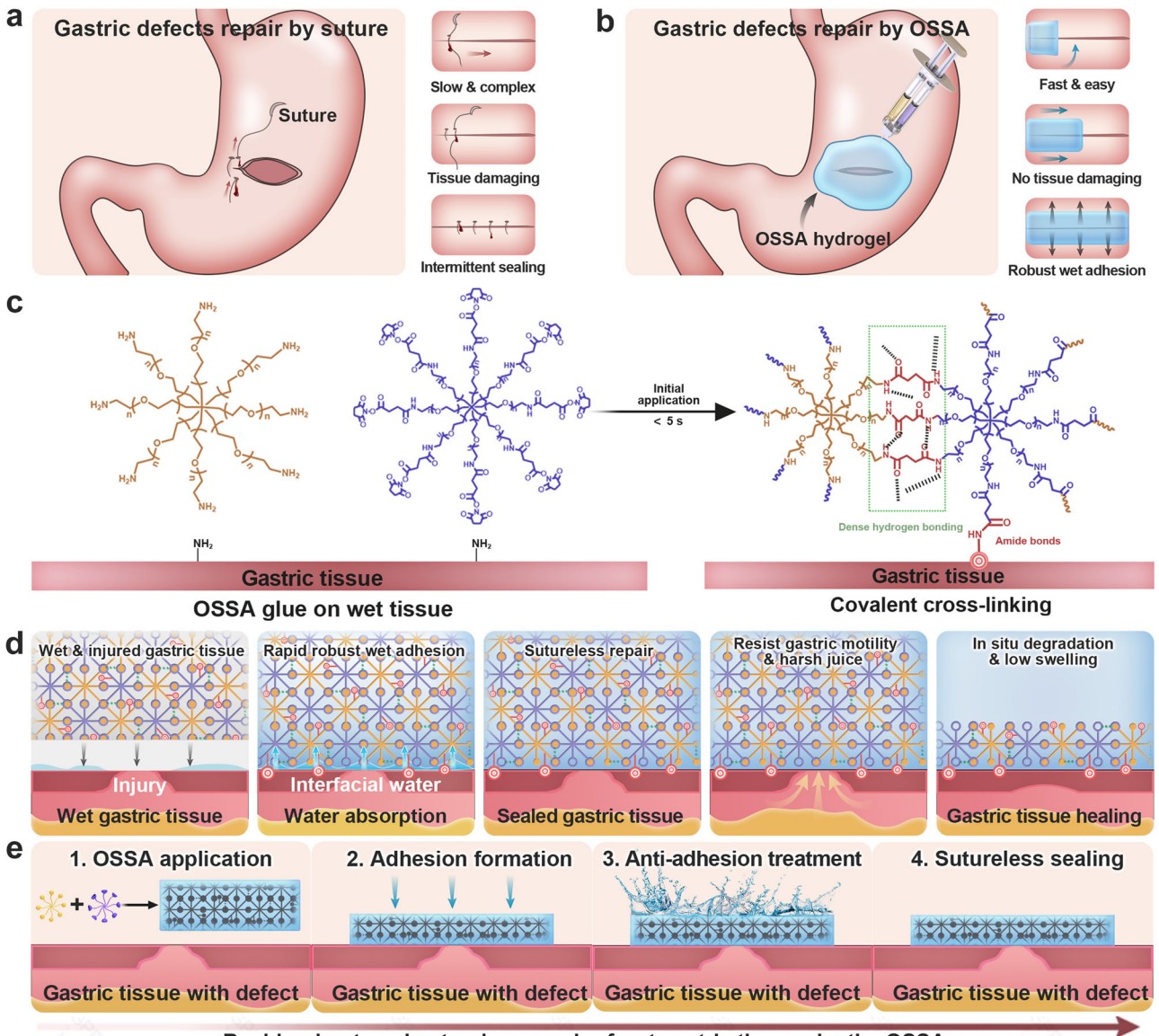

**Fig. 2 | Design and mechanism of sutureless repair of wet gastric tissues by the injectable OSSA hydrogel. a** Schematic illustrations for the repair of gastric defects by sutures. **b** Schematic illustrations for sutureless repair of gastric defects by the OSSA hydrogel. **c** Chemical composition of the OSSA hydrogel based on octa-PEG-SSA and octa-PEG-NH$_2$ polymers and schematic illustrations for rapid wet adhesion onto the gastric tissue surface by covalent cross-linking based on interfacial amide bonds. **d** Schematic illustrations for the mechanism of instant adhesion and long-lasting closure of gastric defects by the OSSA hydrogel based on the dry cross-linking process and well-matched degradation behavior. **e** Schematic illustrations for the mechanism of anti-adhesion treatment of non-contact surfaces.

further indicating the instant wet adhesion and robust sealing interface of the OSSA hydrogel towards the gastrointestinal tissues, even exposure onto the fluidically, chemically and mechanically dynamic environment.

To evaluate the sealing performance of OSSA hydrogel for ex vivo gastric perforation, an incision of ~10 mm was made on the isolated porcine stomach using a razor blade to simulate the gastric perforation (Fig. 4c and Supplementary Movie 3). The OSSA hydrogel was in situ deposited on the damaged tissues, followed by filling the stomach with more than 5 kg of water without any fluid leakage, which was derived from the succinimidyl-active ester moieties tightly bound to amino groups, as well as its reliable bulk strength on the tissue surfaces. Next, we assessed the leakage behavior of the OSSA hydrogel at the defect site. To simulate ex vivo full-thickness gastric defects, a larger incision (~20 mm) was created in an isolated porcine stomach. The injectable OSSA adhesive, pre-stained with blue dye, was deposited in situ onto

the damaged tissue. Notably, no significant glue leakage or other gel-like substance was observed from the defect regions, corroborating its instant adhesion, fluid-tight capacity, and suitability for sutureless sealing of large gastric perforation (Supplementary Movie 4). Afterwards, an incision of ~6 cm was made on the other two porcine stomachs to simulate the gastric perforation, and the sealed stomachs by OSSA hydrogel and Coseal adhesive were filled with red dye-stained medium of pH 2.0 and immersed into PBS solutions to simulate the biofluid environment (Fig. 4d). No liquid leaking was observed for both the OSSA hydrogel and the Coseal in the early stage. Coseal layer drastically swelled within 4 h, but a burst leak occurred at 8 h, indicating the adhesion failure in the acidic environment (Supplementary Movie 5), which may ascribe to the weak adhesive strength and acid-accelerated ester bond degradation. In contrast, OSSA adhesive could remain firm for more than 72 h even when the porcine stomach was putrid in water (Supplementary Movie 6), verifying a fluid-tight sealing

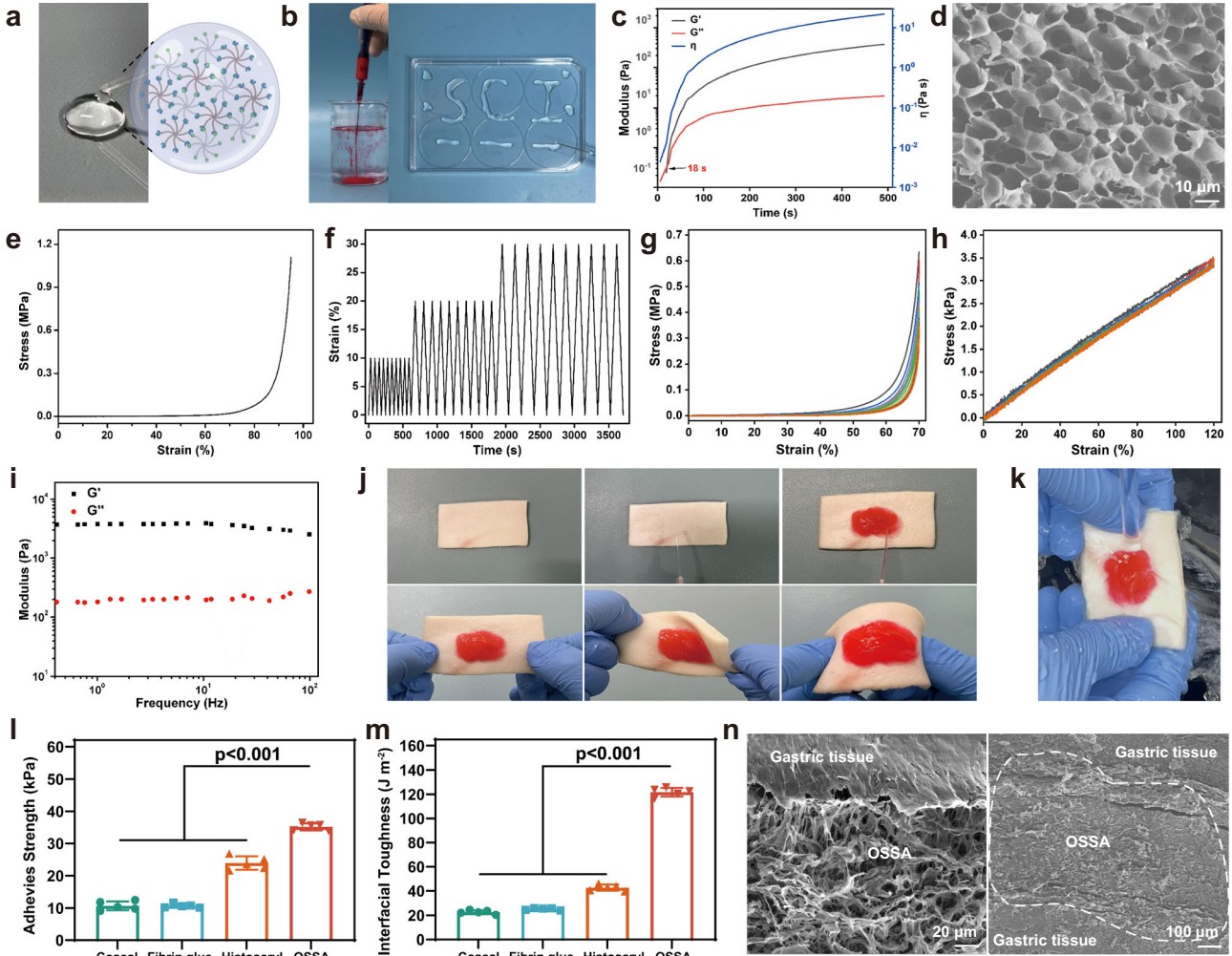

**Fig. 3 | Physicochemical, mechanical, and adhesive performances of the OSSA hydrogel. a, b** Gross appearance of the OSSA hydrogel with well-defined structure and easy injectability. **c** Storage modulus $G'$, loss modulus $G''$ and viscosity $\eta$ of octa-PEG-SSA (8%) and octa-PEG-NH$_2$ (8%) solutions after mixture as a function of time. Oscillatory frequency: 1 rad s$^{-1}$. **d** SEM image of the OSSA hydrogel (all experiments were independently repeated at least 3 times with similar results, scale bar: 10 μm). **e, f** Compressive stress and repeated compression at strains of 10, 20, and 30% of the OSSA hydrogel. **g** Cyclic stress–strain compressive curves at a strain of 70% and (**h**) cyclic stress–strain tensile curves at a strain of 120% of the OSSA hydrogel. **i** Rheological analysis of the OSSA hydrogel. **j, k** Photographs of OSSA hydrogel adhered to porcine skin under different deformations and water resistance measurement. **l** Adhesive strength and (**m**) interfacial toughness of the OSSA hydrogel and the commercially available tissue adhesives on porcine skin. Data are presented as mean ± SEM ($n$ = 5 independent samples). $P$ values were determined by one-way ANOVA with the least significant difference post hoc test (**l, m**). **n** SEM images of the adhesion interface between the OSSA hydrogels and the porcine gastric tissues (all experiments were independently repeated at least 3 times with similar results). The left image is the main view (scale bar: 20 μm), and the right image is the top view (scale bar: 100 μm). Source data are provided as a Source data file.

of the perforated stomach by the acid-tolerant amide linkages and demonstrating a reliable sealing of the gastric defects. Although we did not extend the observation period due to the difficulties in preserving porcine stomach samples in such ambient conditions, we demonstrated that the firm adhesion onto tissues could last longer (>2 weeks) in human gastric juices via a simple in vitro soaking (Supplementary Fig. 5).

### In vitro swelling and elucidation of the underlying mechanism for the performance leap

During the long-term contact with acidic fluids, the hydrogel bioadhesives in vivo may experience undesirable swelling and subsequent gradual deterioration in strength and interfacial robustness. On account of the amide bonds within the compact network, the OSSA hydrogel has unexpected stability to maintain low swelling and withstand strong acid and multi-enzyme erosions. After immersing the OSSA hydrogel into the PBS solution, porcine stomach fluid and human stomach fluid (Supplementary Fig. 6), the swelling ratio of

OSSA hydrogel in PBS and porcine gastric fluid was only 24% in 10 days while it gradually climbed to a plateau of 33.7% originating from the inhalation of a variety of organic and inorganic substances in the human gastric juices (Fig. 4e). Contrary to previously reported ester-linked tetra-PEG hydrogels with high swelling ratio of more than 250%[26,39,41], this low swelling of OSSA hydrogel may significantly ascribe to its stable multi-amide linkages and regionally concentrated hydrogen interactions bonding throughout the dense network (Supplementary Fig. 8), which could significantly hinder the fluid penetration and other soluble substances. Hydrogen bonding interactions significantly affected the infrared characteristic peaks of amide bonds, which were mainly manifested as the red shift and peak shape changes of N–H and C=O stretching vibrations[43,44]. As shown in Fig. 4f, the formation of typical new peak ($v_{C=O}$) at 1678 cm$^{-1}$, widening peak shape ($v_{N-H}$) at 3140–1740 cm$^{-1}$ and obvious red shift collectively verified the intermolecular hydrogen bonding interactions among the stable amide-linked skeleton, compared to the similar structure of OSS hydrogel (Supplementary Fig. 9). To further distinguish the differences

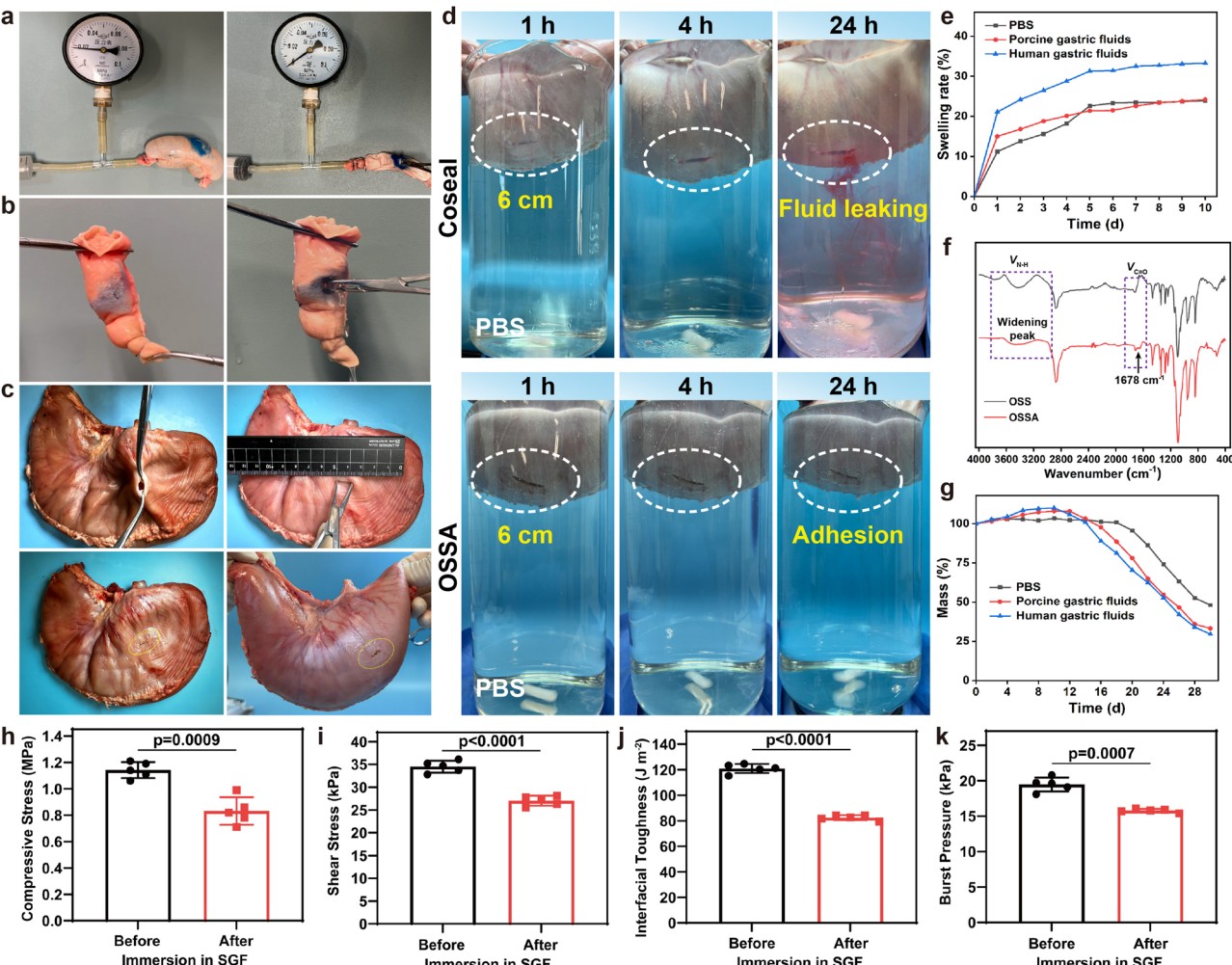

**Fig. 4 | Ex vivo demonstration of the robust sealing effect in gastric perforation with low swelling ratio and slow degradation time. a** Bursting pressures of the OSSA hydrogel towards the damaged porcine colon tissues (diameter: 4 mm). **b** Photographs of the in vitro sealing effects of OSSA hydrogel toward the porcine colon tissues after application and destruction. **c** Photographs of in vitro sealing of porcine stomach (diameter: 10 mm) using the OSSA hydrogel. **d** Photographs of ex vivo sealing of porcine gastric defects (diameter: 6 cm) using the OSSA hydrogel and commercial Coseal adhesive, where the porcine stomach was filled with red dye-stained pH 2.0 solution ex vivo prior to immersion in PBS to demonstrate the acid-tolerant adhesion and sealing stability. **e** Swelling ratios of the OSSA hydrogel in PBS solutions, porcine gastric fluids, and human gastric fluids. **f** IR spectra of OSS and OSSA hydrogels. **g** The degradation profiles of the OSSA hydrogel in PBS solutions, porcine gastric fluids, and human gastric fluids. **h–k** Compressive strength, shear strength, interfacial toughness and burst pressure of the OSSA hydrogel on ex vivo porcine skins before and after immersion in the SGF solutions for 72 h. Data are presented as mean ± SEM ($n = 5$ independent samples). $P$ values were determined by two-sided Student's $t$ test (**h–k**). Source data are provided as a Source data file.

between ester-based PEG network and amide-based PEG network, we systematically compared OSSA hydrogel with previously reported ester-based OSS hydrogel and commercial Coseal sealant under identical harsh conditions (simulated gastric fluid, pH 1.5 with pepsin). Compared with the structural stability and durable wet adhesion of OSSA hydrogel, both OSS hydrogel and Coseal sealant with the same solid contents were completely degraded within 2 days (Supplementary Fig. 10) due to the rapid hydrolysis of ester groups in such harsh SGF conditions, demonstrating interfacial adhesion failure in vitro and inferring the impossibility on sealing gastric perforations under the acidic fluids in vivo for a long period. Monitoring the degradation in acid found that the ester-based OSS hydrogel rapidly broke down via ester hydrolysis, releasing soluble PEG fragments. The entirely disappeared characteristic peak of $CH_2$ groups (*a*) next to the oxygen atom of ester groups in $^1$H NMR spectra clearly pronounced the complete fracture of the ester bonds (Supplementary Fig. 11), indicative of the acid-induced degradation of the OSS hydrogel. In contrast, the OSSA bioadhesives could remain intact with no detectable soluble degradation products over the same period. Therefore, these stable

amide linkages and locally dense hydrogen bonds throughout the compact framework endowed OSSA hydrogel with tolerance on acidic and multi-enzyme environments. Under this circumstance, the adhesive performance was slightly decreased when the OSSA hydrogel was fully swollen in the simulated gastric fluid (SGF, pH 2.0) solution after initial adhesion onto ex vivo porcine skins for 72 h, but the decreased strength was not statistically significant, originating from its low swelling ratio and weak interfacial deterioration. The fully swollen OSSA hydrogel could still maintain superior adhesive performance with high interfacial toughness (>85 J m$^{-2}$), shear strength (>26.5 kPa), compressive strength (>0.8 MPa) and burst pressure (>15.6 kPa) in comparison with commercially available tissue adhesives while adhesively contacting with wet tissues (Fig. 4h–k). Additionally, the OSSA hydrogel in gastric juices exhibited a slower degradation process with less than 5% weight loss within 2 week and the residual mass was still more than 25% over 1 month (Fig. 4g). It was mentioned that initial mass increase was attributed to inhalation of various components in the mediums, and acidic and digestive enzymes finally induced the hydrolysis or degradation after 2 weeks, indicating the reliable acid-

tolerance for gastric perforation repair and compatible degradation with newborn tissue regeneration in vivo. This low swelling and slow degradation, in conjunction with the durable wet adhesion described above in gastric juices, explicitly verified its long-term gastric resistance to deterioration in strength and interfacial robustness.

To gain atomic-level insight into the crosslinking network and elucidate the underlying mechanism for this performance leap, we then construct simplified molecular models to probe the non-covalent interactions within the crosslinked networks at the atomic level. The models were designed to isolate the effect of the crosslinking chemistry: both models featured an identical octa-armed PEG core but were connected via either ester and amide linkages (OSS network) or only amide linkages (OSSA network), respectively. The quantum theory of atoms-in-molecules (QTAIM) analysis provided the direct evidence that the amide linkages in the OSSA network fostered a superior hydrogen-bonding network compared to the ester linkages in the OSS network (Supplementary Fig. 12). For the characteristic N–H···O hydrogen bond interaction in the amide-linked Network, the average electron density ($\rho$) at the so-called bond critical point (BCP) was 0.025814 a.u., corresponding to a medium-strength hydrogen bond. In contrast, the strongest comparable interactions in the ester-linked network exhibited a significantly lower average electron density of 0.022932 a.u. Using the empirical correlation of $E\_HB = -223.08 \times \rho(BCP) + 0.7423$[45], the average hydrogen bond energy in the OSSA network was −21.95 kJ/mol, which was approximately 43% stronger than the strongest interactions found in the OSS network (−15.37 kJ/mol). This indicated that not only were there more hydrogen bonds in OSSA, but also the average strength of these hydrogen bonds was higher. In addition to strong N–H···O hydrogen bonds, we also observed the weaker C–H···O interactions. The slightly lower energy of these interactions in OSSA (−3.12 kJ/mol vs. −4.13 kJ/mol in OSS) was consistent with a more crowded molecular environment where the dominant strong amide-amide hydrogen bonds optimize the packing, leaving less configurations for the weaker C–H···O interactions to form optimally. This overall shift towards a higher proportion of strongly cooperative hydrogen bonds within the dense OSSA network contributed to a more rigid and compact structure.

The computational analysis results unequivocally demonstrated that the simple change from an ester to an amide linkage in an otherwise identical octa-PEG architecture fundamentally altered the hydrogen bonding landscape. The amide groups in the OSSA network served as potent sites for forming strong and cooperative N–H···O hydrogen bonds, creating a robust and dense network of non-covalent crosslinks. This provided the atomic-level mechanism for the observed macroscopic superiority: the synergistic effect of covalent crosslinks and robustly dense hydrogen-bonding network severely restricted the polymer chain mobility and created a formidable barrier against the penetration of water and hydronium ions (e. g., $H_3O^+$, enzyme). This directly explained the exceptionally low swelling ratio and good stability of OSSA hydrogel in the acidic gastric environment, where the ester-based OSS or Coseal network would be rapidly hydrolyzed. Therefore, this fundamental structure-property relationship, clarified from the electronic to the macroscopic level, validated our molecular design strategy and highlighted its transformative potential.

### In vitro degradability and biocompatibility of the OSSA hydrogel

To further assess in vivo degradation behaviors of bioadhesives, Cy5.5-labeled OSSA hydrogel was surgically performed by subcutaneous implantation in rats. The degradation processes were monitored by in vivo bioluminescent imaging, histology, and immunofluorescence examinations. The morphological observation and bioluminescent detection showed that OSSA hydrogel adhered onto subcutaneous tissues was effectively degraded after 8 weeks (Fig. 5a–c), which was far enough to satisfy the time required for gastric healing. H&E, Masson,

and immunofluorescent staining also confirmed this degradation behavior of OSSA hydrogel over time. When the OSSA hydrogel was subcutaneously implanted at day 3, it elicited no significant inflammatory response that was consistent with the clinically approved Coseal, indicating the favorable host immune response of OSSA hydrogel with clear verification of histological and immunofluorescence examinations (Supplementary Fig. 13). At 8 weeks, the histological features returned to normal, with no typical inflammatory responses and fibrosis (Fig. 5d). Furthermore, on account of the significant differences in tissue types and metabolic characteristics between the subcutaneous and enterocoelia, we further investigated the residual volumes of OSSA hydrogel adhering onto the rat omentum majus to verify its intra-abdominal degradation. The results displayed that the degradation rate of OSSA hydrogel was significantly faster than that of subcutaneous tissue, and OSSA hydrogel could not be detected by bioluminescence imaging within 4 weeks (Supplementary Fig. 14). Even so, the degradation time of the OSSA hydrogel could meet the mechanical closure requirement during the healing process of gastric perforation (>2 weeks). CD3, CD68 and iNOS serve as biomarkers of activities of T cells and macrophages, while the concentrations of collagen I and collagen III reflect fibrosis progression. In situ expression of these above markers was downregulated in subcutaneous tissues (Fig. 5e and Supplementary Fig. 15), suggesting that the immune response and fibrosis were naturally weakened without persistent immune rejection, which reaffirmed the high biocompatibility and in vivo degradability of the OSSA hydrogel.

The medical application necessitates biocompatibility of hydrogel bioadhesives, and we thus conducted live/dead, CCK-8, and cell apoptosis assays to evaluate the long-term biocompatibility of OSSA hydrogel. Human gastric mucosa epithelial cell line (GES-1) was co-cultivated with commercial Coseal adhesive, OSSA hydrogel and control medium. Compared to the Coseal group, the OSSA group exhibited higher cell viability and lower cell apoptosis of GES-1, analogous to the control group (Fig. 5f, g, and Supplementary Fig. 16), indicating the insignificant impact of OSSA on the maintenance and proliferation of gastric mucosa epithelial cells. Additionally, no obvious differences in cell densities and morphologies were observed between the OSSA and the control groups, as well as almost no dead cells (red fluorescence) were detected, further validating that OSSA hydrogel had high biosafety on survival and functions of stomach epithelial cells in vitro. To further monitor the overall health of animals during the hydrogel subcutaneous implantation period, the complete blood counts, blood biochemical indexes and IL-6 and TNF-α cytokines were determined at different time to reflect the biocompatibility in vivo. During the hydrogel degradation, there was no significant difference in blood analysis among the suture, Coseal and OSSA groups, indicating no systemic toxicity of the degradation byproducts (Fig. 5h–k). Collectively, these findings supported satisfactory biodegradability of OSSA bioadhesive to resist the gastric fluid erosion environment and biocompatibility for gastric perforation repair.

### In vivo gastric adhesion and sealing efficiency in rat models

Having demonstrated the unique acid-tolerance, ex vivo instant and tough adhesion to gastric tissues, in vitro long-term robustness, and slow biodegradability of OSSA bioadhesive, we then investigated its sutureless repair of gastric perforation and healing efficacy with the suture and Coseal as controls in rat models (Fig. 6a). An 8-mm hole was created on the surface of the small stomach and the hydrogel precursor solutions were then injected into the wound site (Fig. 6b). On the basis of the quick gelation and tight adhesion onto tissues, the OSSA bioadhesive could be easily sprayed on the injured sites within a few seconds to provide robust sutureless sealing, free from the dependency on additional preparation steps and operation devices like coagulation inductions, ingredient blending and material trim. Furthermore, the adhesion of noncontact tissue surface could be

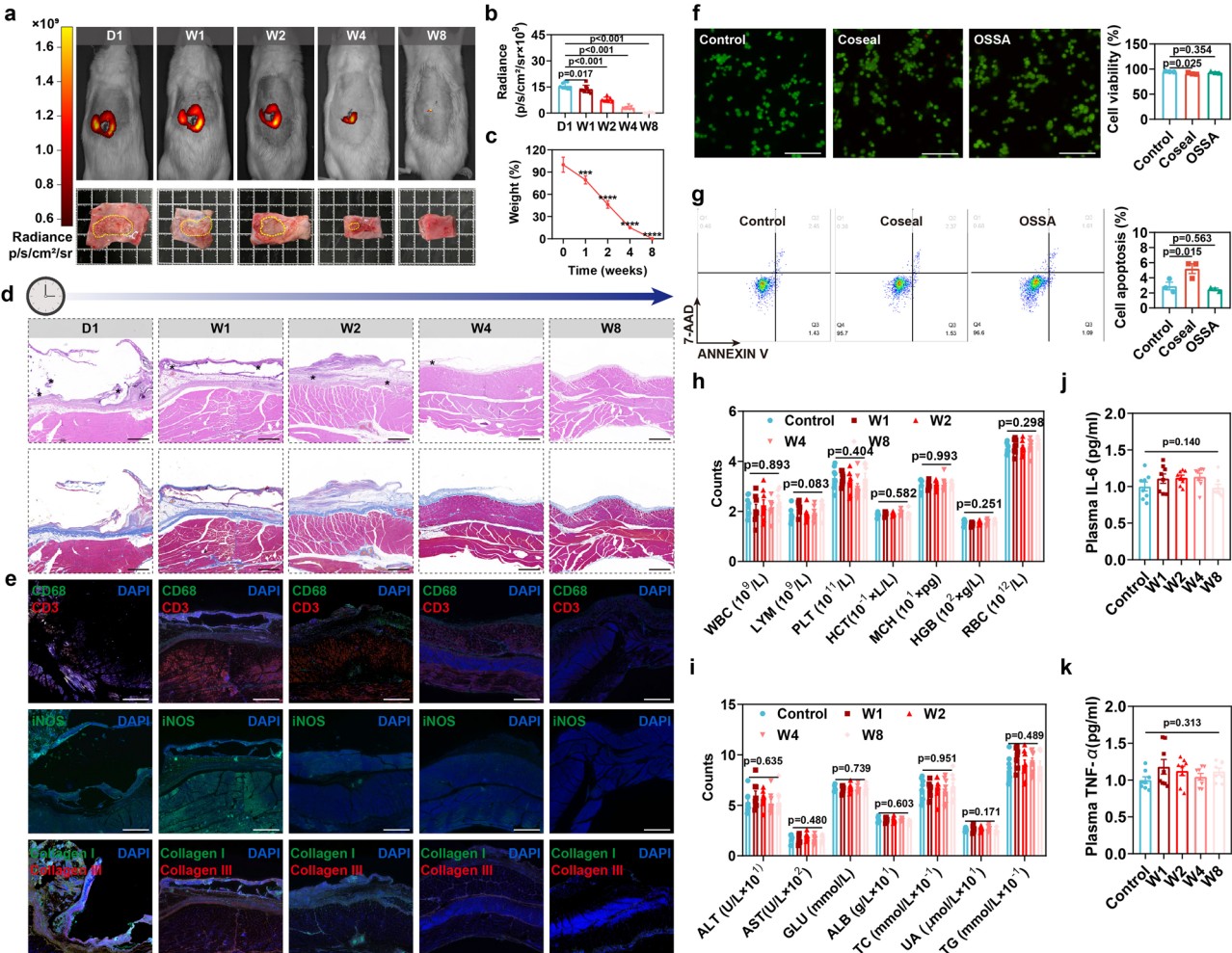

**Fig. 5 | In vitro and in vivo biodegradation and biocompatibility of the OSSA hydrogel. a** Representative bioluminescent (upper) and morphological (lower) images showing the subcutaneously implanted OSSA hydrogel at day 1, week 1, 2, 4, and 8, respectively. Data are presented as mean ± SEM (*n* = 8 independent samples). **b** The radiance intensities of residual OSSA hydrogel. Data are presented as mean ± SEM (*n* = 8 independent samples). **c** The remaining weight of residual OSSA hydrogel. Data are presented as mean ± SEM (*n* = 8 independent samples). **d** H&E and Masson staining for the subcutaneous tissues attached with implanted OSSA hydrogel. Scale bar: 500 μm. * Represents the remaining materials. **e** Representative images of immunofluorescence of the tissues marked with CD68, CD3, iNOS, collagen I and collagen III. Scale bar: 500 μm. Three areas per tissues were randomly selected and analyzed. **f** Representative live/dead assay images (left) and the cell viability (right) of GES-1 cells for the control (DMEM), Coseal, and

OSSA hydrogel extracts after 72-h culture. Data are presented as mean ± SEM (*n* = 3 independent samples). Three contemporary measurements from the same experimental batch were performed for the average. Scale bar: 50 μm. **g** Representative flow cytometry (left) and cell apoptosis (right) showing cell apoptotic rates. Data are presented as mean ± SEM (*n* = 3 independent samples). Three contemporary measurements from the same experimental batch were performed for the average. **h–k** Blood routine, biochemistry index, IL-6 and TNF-α of rats with subcutaneous implantation of OSSA hydrogel at day 1, week 1, 2, 4, and 8, respectively. Data are presented as mean ± SEM (*n* = 8 independent samples). The blood sample derived from one rat is regarded as an independent unit. *P* values were determined by one-way ANOVA with the least significant difference post hoc test (**b**, **c**, **f**–**k**). Source data are provided as a Source data file.

rapidly eliminated in vivo after simply rinsing with water or normal saline to accelerate the hydrolysis of residual active ester and completely failure the adhesive groups. This unique feature of OSSA bioadhesive could effectively avoid the harmful postoperative adhesion to other tissues in vivo, and in particular, dramatically reduce the possibility of abdominal adhesion after closure by sutures or other adhesives. In contrast, the suture group required more than 3 min of manual manipulation even for a skilled surgeon and was prone to result in puncture-driven tissue damage. While the longer gelation time (more than 20 s) made it possible for the Coseal group to adhere to other tissues, but its manipulation time and adhesive strength were significantly weaker than that of OSSA bioadhesive in actual use (Fig. 6b and Supplementary Movie 7). Therefore, this combination of rapid adhesion to the injured tissue and avoidably postoperative

adhesion to other tissues made the injectable OSSA hydrogel promising to develop into a convenient double-sided bioadhesive.

Histological and immunofluorescent examinations were then conducted to evaluate the repair process of gastric defects. After 4 weeks, the suture group induced typical fibrosis and inflammation at the injured sites of the stomach along with the obvious suture granuloma and scars, while the Coseal group also exhibited an apparently inflammatory response, which was attributed to the high swelling-induced severe postsurgical adhesions with the neighboring tissues. For comparison, the defects were sealed well without observable surgical complications and inflammatory cells to the underlying and surrounding tissues for OSSA group, and the histological morphologies of stomach tissues were comparable to the normal (Fig. 6c and Supplementary Fig. 17). This was mainly due to the low swelling behavior, nonviscous surrounding tissues, and gradual degradation of

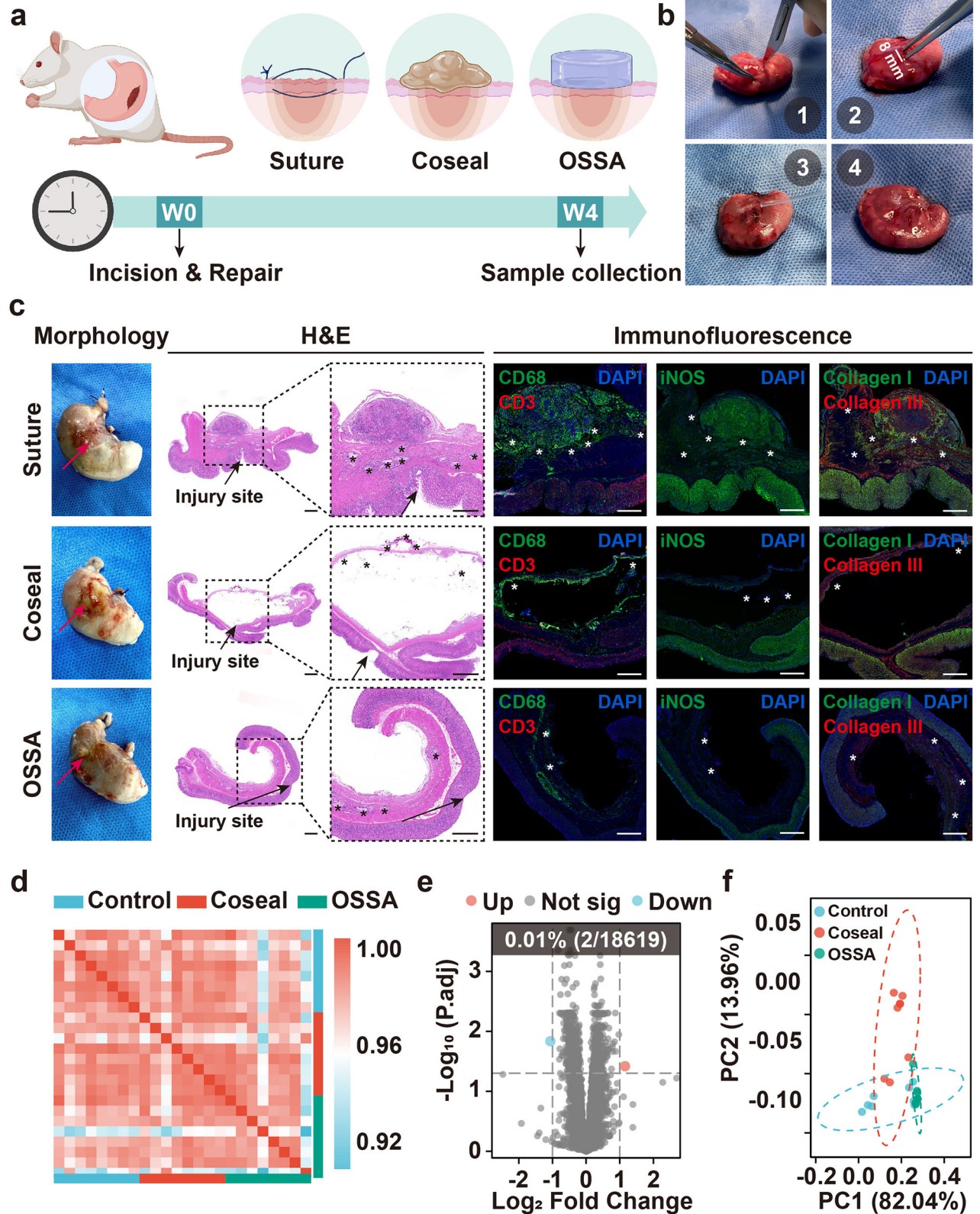

OSSA bioadhesive to avoid the postoperative adhesion or other related side effects. In agreement with the histological analysis, the immunohistochemical staining was performed to verify the lower immune response and fibrosis during the healing process. The OSSA group exhibited lower levels of CD68, CD3, iNOS, collagen I and collagen III, confirming the alleviated inflammatory response, mild fibrosis, and promoted gastric perforation repair (Fig. 6c and

Supplementary Fig. 18). In addition, the blood analysis showed no statistical differences in complete blood routine counts, biochemical indexes, IL-6 and TNF-α among the three groups compared to the healthy rats (Supplementary Fig. 19). Meanwhile, the other organs (liver, kidney, brain, lung, and spleen) also exhibited normal pathological morphologies after treatment of three modalities (Supplementary Fig. 20). Overall, the above results validated our hypothesis that

**Fig. 6 | In vivo adhesion ability of the OSSA bioadhesive in rat models.**
**a** Schematic illustration of gastric defects treated with the suture, Coseal and OSSA bioadhesive in rat models. Created in BioRender. Ze, W. (2026) https://BioRender. com/xj4tsuc. **b** Representative images of rat models with gastric defects closure by OSSA bioadhesive. The steps were marked as follows: (1) complete exposure of the stomach, (2) preparation of the 8-mm incision, (3) injection of the OSSA bioadhesive, and (4) sutureless sealing effect. **c** Gross observation, H&E and immunofluorescent staining for stomach sealed by the suture, Coseal and OSSA bioadhesive after 4 weeks, respectively. The injured sites were labeled with black arrows in the morphological images. Scale bar: 500 μm. * Represents the remaining materials.

Three areas per tissues were randomly selected and analyzed. **d** Correlation analysis of RNA-seq data of stomach epithelium treated with the Coseal, OSSA bioadhesive and control healthy tissues (*n* = 8 independent samples). **e** The volcano plot displaying the differentially expressed genes (*P* < 0.05, fold change > 1 or < −1) in the OSSA group compared to the control group. The percentage and number of the differentially expressed genes in all detected genes were indicated in white (*n* = 8 independent samples). **f** The principal coordinate analysis comparing β diversities of gastric microbiota of rats in the control, Coseal and OSSA groups. Source data are provided as a Source data file.

OSSA bioadhesive with reduced inflammatory response and negligible systemic toxicity could efficiently achieve the sutureless repair of gastric perforation without the aid of additional staples or sutures in a preclinical animal model.

Morphology observations and basic biomarker concentration determination could only reveal the basic physio-pathological process of gastric defect closure, but the underlying mechanisms remained a black box. To deeply investigate the impact of OSSA bioadhesive on the gastric homeostasis, we creatively depicted the alterations in transcriptome and microbiome using RNA-seq and 16s rRNA sequencing. For the gastric defect models, transcriptome patterns had high similarities with correlation coefficients exceeding 0.9 (Fig. 6d). Differentially expressed genes (*P* < 0.05, $Log_2$ fold change > 1 or < −1) accounted for only 0.01% in the OSSA group, which was typically smaller than that in the Coseal group (Fig. 6e and Supplementary Fig. 21a). The principal component analysis of RNA-seq data showed no significant differences in the transcriptome between the OSSA bioadhesive and suture group, whereas Coseal group induced greater alterations in the landscape of gastric gene expression (Supplementary Fig. 21b). Furthermore, the balance of gastric bacteria was closely associated with stomach functions. The α diversity based on 16s rRNA sequencing data indicated that OSSA bioadhesive had no significant effect on the homogeneity of gastric bacterial communities, whereas Coseal downregulated α diversities (Supplementary Figs. 21c–e). Principal coordinate analysis showed that interference of Coseal on β diversities were also more significant than that of OSSA bioadhesive (Fig. 6f). Taken together, the high-throughput results revealed the negligible interference on gastric transcriptome and microbiome, also reflecting the biosafety and clinical application prospect of OSSA bioadhesive. All these results indicated that the OSSA bioadhesive could be efficiently used for the sutureless sealing and gastric perforation repair without causing long-term inflammatory response and gastric transcriptome and microbiome interferences.

To gain deeper insights into the potential molecular mechanisms underlying OSSA hydrogel-induced gastric tissue regeneration, we performed transcriptomic analysis on the repaired gastric tissue and compared it with the control group. As shown in the molecular mechanism diagram, differentially expressed genes between OSSA bioadhesive and the control group were analyzed by gene ontology (GO) analysis using the clusterProfiler hypergeometric distribution algorithm to obtain enrichment results in three categories: molecular function, biological process, and cellular component. The analysis revealed that upregulated genes potentially directly related to gastric tissue healing included muscle contraction, actin binding, cytoskeletal motor activity, and myofibril assembly (Supplementary Fig. 22a). Further GO enrichment and KEGG pathway analysis indicated that OSSA bioadhesive modulated the inflammation, fibroblast activation, angiogenesis, macrophage activity, and clearance of apoptotic cells to promote inflammation resolution. Additionally, OSSA bioadhesive could drive cytoskeletal movement, mediate epithelial cell migration, and promote tissue regeneration through mechanisms such as cytokine–cytokine receptor interaction, efferocytosis, and motor proteins signaling pathways, thereby exerting its regulatory effects on wound healing (Supplementary Fig. 22b).

## Sutureless repair of the gastric perforation in porcine models
We then compared the repair efficacy of OSSA hydrogel, surgical suture, and two commercially available tissue adhesives (Coseal and Histoacryl) in the large gastric perforation of porcine models using the minimally invasive technique (Fig. 7a). A full-thickness defect of approximately 3 cm was created by the laparoscopic electrosurgical knife in the anterior wall of the greater curvature of the pig's stomach near the pylorus while simultaneously performing an endoscopic observation of the gastric mucosal side (Supplementary Movie 8). The sealing by the suture and bioadhesive was administered via the laparoscopic route. The OSSA, Coseal, and Histoacryl sealants could be facilely delivered and robustly adhered onto the slippery gastric defects within a short time, while the suture sealing procedure took more than 15 min. There were no observable leakages of gastric contents and juices after treatment of OSSA, Coseal, Histoacryl, and suture sealing. Throughout the follow-up period, all treated porcine in the OSSA and suture groups survived with normal feeding behavior and steady weight gain; additionally, there were no signs of abnormal health conditions (e.g., fever or lethargy) or complications associated with wound healing based on daily veterinarian monitoring during the period of 4 weeks, revealing good postoperative recovery. In contrast, porcine treated with Coseal and Histoacryl adhesives exhibited significant weight loss, likely attributable to the inadequate defect repair, severe abdominal adhesions, and secondary infections (Fig. 7b). The dynamically monitoring results from the portable electrocardiograph scanner showed that cardiovascular functions of these pigs remained normal after modeling (Supplementary Fig. 23) without occurrence of abnormal clinical symptoms of fever, increased heart rate, heart rhythm abnormality, restlessness, and lethargy.

The stomach is a multi-layer hollow organ, and its full-thickness recovery requires comprehensive evaluation. The combined laparoscopic-endoscopic technique in the clinic can give full play to the respective advantages of endoscopy and laparoscopic, which is of great significance for clinical evaluation of gastric perforation repair based on the multi-layer morphologies of both the serosal and epithelial surfaces. Thus, combined laparoscopic-endoscopic surveillance was performed every week to assess the healing process and repair efficacy (Fig. 7c). The observation results showed that the OSSA bioadhesive was gradually degraded and the mucosal surface of the stomach gradually healed with the passage of postoperative time (Supplementary Movies 9 and 10). It was mentioned that the clear leakages of gastric contents and peritonitis at the early stage in the laparoscopic images reflected the establishment of large-scale porcine gastric perforation models. The healing speed was higher with insignificant scar, peripheral mucosal edema, and inflammatory response compared to the suture group (Fig. 7d). To assess the risk of postoperative adhesions, we employed the validated modified American Fertility Society scoring system and conducted blind laparoscopic examinations in porcine models (Supplementary Fig. 24). Two independent investigators quantitatively assessed postoperative adhesion through the intensity, scope, and affected area metrics. It was noteworthy that OSSA bioadhesive not only achieved the superior repair efficacy comparable to the clinical suture method in terms of gastric defect repair, but also had significantly lower levels of abdominal

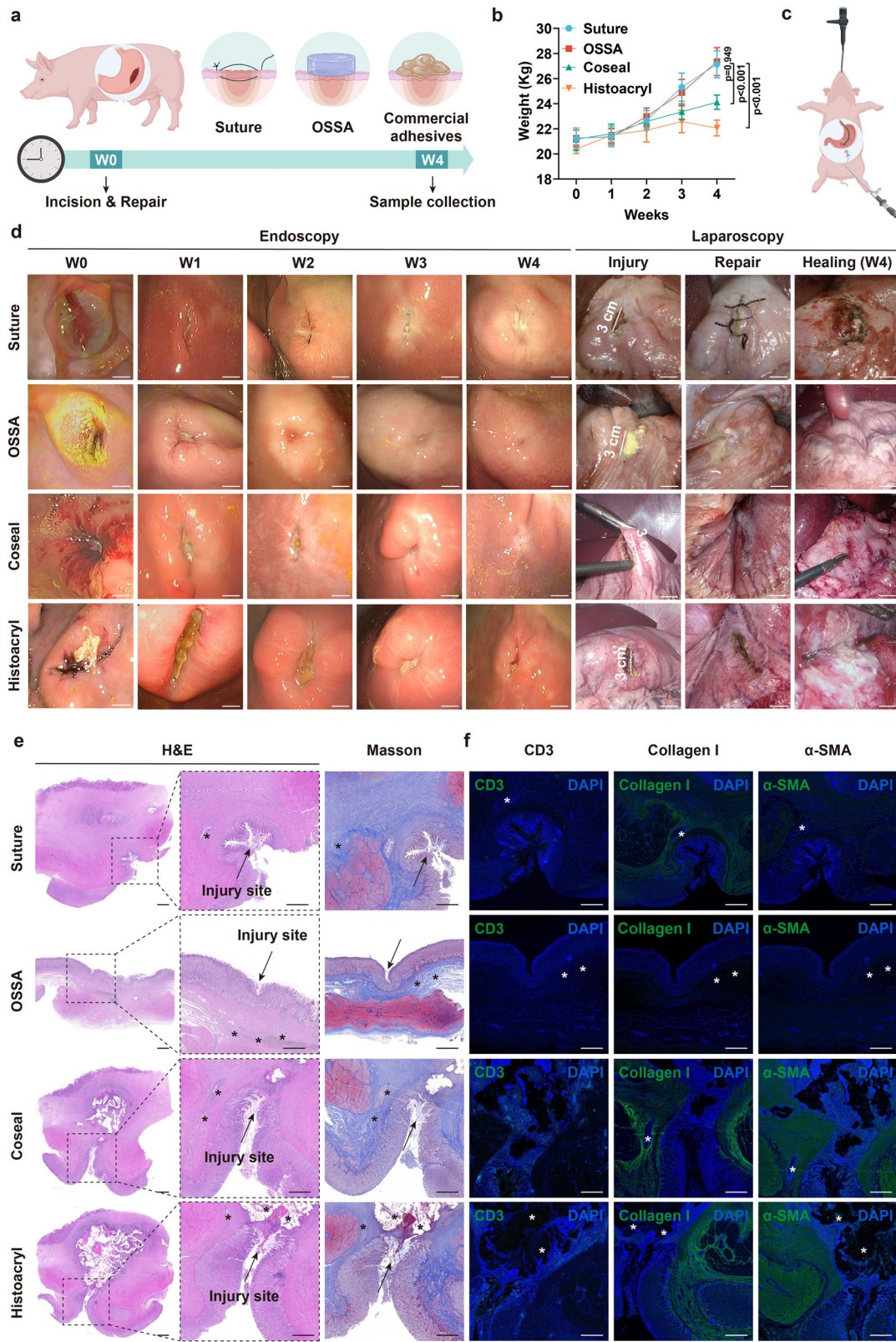

adhesions. Compared to the control groups with varying degrees of severe adhesions and organ displacement, the OSSA group had the lowest score for abdominal adhesions. This anti-adhesion performance had a dual function on effectively preventing pathological adhesions at the repair sites and preserving the integrity of the surrounding tissues, which offered a significant and innovative advantage over existing solutions. Therefore, this double-sided or Janus-like OSSA bioadhesive

had significant potential in gastrointestinal repair and postoperative adhesion prevention.

In addition, H&E and Masson staining confirmed the defects repaired by the OSSA exhibited less fibrosis around the defects after 4 weeks without obvious inflammation, abscess, and granuloma, which validated that OSSA bioadhesive could achieve the leak-free and sutureless healing of porcine gastric defects, displaying a more

**Fig. 7 | Efficiency of OSSA bioadhesive in sealing large gastric defects in porcine models. a** Schematic illustration of gastric defects treated with the suture, OSSA, and commercial adhesives in porcine models. Created in BioRender. Ze, W. (2026) https://BioRender.com/xj4tsuc. **b** Weight changes of pigs with gastric defects after 4 weeks of treatment with the suture, OSSA, Coseal, and Histoacryl groups. Data are presented as mean ± SEM ($n = 5$ independent samples). The weight value from one pig at the indicated time point is regarded as an independent unit. $P$ values were determined by two-way ANOVA with the least significant difference post hoc test. **c** Schematic illustration of the combined laparoscopic-endoscopic technique for

monitoring the healing process and repair efficacy. Created in BioRender. Ze, W. (2026) https://BioRender.com/xj4tsuc. **d** Representative endoscopic and laparoscopic images of the porcine stomach at various periods ($n = 5$ independent samples). **e**, **f** H&E, Masson and immunofluorescence of the tissues marked with CD3, collagen I, and α-SMA staining of porcine gastric defects treated with the suture, OSSA, Coseal and Histoacryl groups for 4 weeks ($n = 5$ independent samples). Scale bar: 500 μm. * Represents the remaining materials. Source data are provided as a Source data file.

histologically favorable outcome (Fig. 7e). Similarly, immunofluorescence staining and normalized immunofluorescence intensity analysis also displayed that the macrophage markers of collagen I, T cells (CD3), and α-SMA were all lower than those in the suture, Coseal, and Histoacryl groups (Fig. 7f and Supplementary Fig. 25), further indicating a lower degree of fibrosis and inflammatory response in the OSSA bioadhesive during the long-term healing process, in agreement with the fibrosis and inflammation observed in the histological evaluation. Besides, the indexes of blood routines, biochemistry, IL-6 and TNF-α concentrations were all within the normal range (Supplementary Fig. 26). Collectively, these findings supported by the combined laparoscopic-endoscopic surveillance, histological evaluation and immunofluorescence examinations verified that the OSSA bioadhesive could offer leak-free sutureless repair with significant superiority than surgical suture and commercially available tissue adhesives for the large size defect model of porcine stomach.

## Discussion

In the past decades, the clinical and preclinical needs have motivated tremendous efforts to develop bioadhesive materials for emergency hemostasis and wound closure, which feature superior flexibility, robust tissue adhesion, good biodegradability and biocompatibility with the wet and dynamic biological environments[16,37,39,46]. However, many existing adhesives and sealants may experience undesirable swelling and subsequent gradual deterioration in strength and interfacial robustness in complex clinical settings, especially during long-term contact with acidic fluids and digestive enzymes for the repair of gastric damages. In addition, the irregular surfaces and dynamic motion of gastric tissues may cause the interfacial fatigue failure, thus posing a significant obstacle to its application as a minimally invasive technique[47,48]. Moreover, how to combine the robust acid-tolerant adhesion with appropriate biodegradability to effectively avoid adhesive failure, fluidic leakage and postoperative adhesion remains a critical challenge in clinical applications. Overall, the limitations of existing technologies highlight the unmet clinical requirements and the vital necessities on developing treatment options for gastric defect repair.

Here, we developed an injectable OSSA bioadhesive for atraumatic, facile and sutureless repair of gastric defects. Its unique attributes of ultrafast gelation, instant wet adhesion, mechanical ruggedness, acid-tolerance, and preferable biodegradation enabled to meet the complex and stringent requirements for sealing large-size of gastric defects. Given the ease of in situ injection and instant adhesion to tissues, OSSA hydrogel could be simply delivered and robustly adhered into the porcine gastric defect sites through laparoscopic equipment as a minimally invasive technique, thus providing a protective sealant regardless of the excessive gastric motility, acidic pH and harsh biofluidic environment. More importantly, compared to previous studies in which adhesive materials were immersed in a single acid-resistant condition or simulated gastric juices in vitro, our OSSA sealant with stable amide-linked skeleton and strong hydrogen bonding interactions within the dense crosslinking network was able to withstand actual porcine and human gastric juices containing digestive enzymes, intrinsic factors, and mucinous proteins. Thus, the laparoscopically assisted in situ injection strategy could provide surgeons

with more control over the treatment process and minimal trauma to patients during current clinical procedures.

In analogous to the commercially available tissue adhesive of Coseal, OSSA hydrogel was also composed of FDA-approved PEG components, indicative of its biocompatibility and potential clinical applications. In this work, we used the surgical suture method and two commercially available commercial adhesives (Coseal and Histoacryl) as controls to validate the sealing efficacy of OSSA bioadhesives. The effectiveness of the suture depended heavily on surgical skills, assistive equipment, and anatomical location. Its operation time was significantly longer than that of adhesive shielding. For the Coseal adhesive, it exhibited higher swelling behaviors despite the considerable sealing effect, so there was a great deal of residual adhesives and postoperative organ adhesion in the commissure and abdominal cavity. While the Histoacryl adhesive could achieve rapid sealing of gastric defects compared to other closure methods, it demonstrated poor local tissue compatibility. The adhesive-tissue interface exhibited significant loss of native elasticity and toughness (Fig. 7d), indicating the potential cytotoxic effects. Through a series of endoscopic and laparoscopic evaluations, weight monitoring and postoperative histological analysis, it was found that the Histoacryl group had incomplete tissue regeneration, sustained inflammatory responses, and severe postoperative adhesions, which even led to the organ displacement. These findings collectively indicated that the commercially available tissue adhesives were not suitable for the repair of large gastric perforation. While the healing effect of OSSA bioadhesive on the injured site was significantly better than that of the suture, Coseal and Histoacryl groups, embodied by normal pathological characteristics and less fibrosis and inflammation. More importantly, on account of the feasible adhesion failure of noncontact surface in vivo by rinsing with water or normal saline to eliminate the residual active ester, the wounds were favorably repaired with newly smooth tissue surface without visible postoperative adhesion and other related side effects. This double-sided or Janus-like feature not only enabled the OSSA bioadhesive with the efficient closure of gastric perforation under the harsh environment but also allowed little damage on surrounding tissues without affecting the normal functions of the organ.

The underlying mechanism of this exceptional performance originates from the synergistic effect of two key design elements: chemically stable backbone and dense hydrogen bonding. On the one hand, replacing easily hydrolyzable ester linkages with stable amide linkages provided intrinsic resistance to acidic and enzymatic hydrolysis. On the other hand, the densely crosslinked network with concentrated hydrogen bonding in octa-armed architecture enabled the formation of a much more densely crosslinked network upon gelation. Computational QTAIM analysis further revealed that octa-armed amide-linked design fosters a dense network of strong hydrogen bonds, providing the atomic-level rationale for its exceptional stability in the harsh gastric environment. This dense network, fortified by regionally concentrated hydrogen bonding, acts as a formidable barrier against gastric fluid penetration and chain disentanglement, leading to exceptionally lower swelling and slower degradation.

A combination of the transcriptome and microbiome had been used to evaluate the superiority of adhesives for sutureless repair of gastric defects. Most of the existing studies only used the

morphological and histological indicators to evaluate the safety and efficacy. However, these indicators were incapable of reflecting the landscape of adhesive effects, and the intuitively unmeasurable alterations were likely to be ignored. Traditional laboratory indicators, such as blood routine, biochemical and plasma inflammatory factors, could dynamically exhibit the systemic changes, whereas have predominant weakness in revealing local pathology and organ homeostasis. Our previous findings demonstrated that physical damage to the gastrointestinal tract induced significant disruption of bacterial community and gene expression signatures[49,50], which inspired us to use transcriptome and microbiome as important indicators of gastric sealing efficacy. The results showed that OSSA bioadhesive caused minor interference with the gastric transcriptome and microbiome, significantly greater than that of Coseal sealant. This suggested that the molecular landscape may be more sensitive to damage and subsequent repair than the classical phenotype, as well as more suitable for investigating the subtle and long-term adhesive effects. These high-throughput results provided insights for evaluating the repair efficiency of tissue bioadhesives.

Prior to clinical trials, investigation on large animals with larger perforation was further necessitated to assess the in vivo adhesion and sealing performance to determine the applicability and efficacy. Here, it was worth noting that our findings proved that the OSSA bioadhesive was capable of sealing injury of ex vivo stomach of 6 cm incision and robust wet adhesion of in vivo porcine gastric perforation of 3 cm, far exceeding the currently reported gastric defect models. In addition, OSSA delivery relied on the laparoscopic means while the instant closure of gastric defects and intuitive repair efficacy were weekly monitored by a combined laparoscopic-endoscopic technique, which provided more conclusive evidences on instant wet adhesion and clear observation of the healing process for potential clinical applications. With all the above-mentioned fascinating traits, we believe that the eminent OSSA has great potentials as a next-generation bioadhesive for clinical repair of gastrointestinal defects.

Nevertheless, our study also revealed some limitations and areas for future works. First, although the OSSA bioadhesive provided facile sutureless sealing of gastric defects at the indicated sites, further validation and better optimization for more geometrically and anatomically complex defects are required. Its applicability to other gastric locations, organs and anatomic environments adjacent more complicated tissues needs further exploration. Second, comprehensive in-vitro and in vivo trials evaluated the short-term safety and efficacy of OSSA bioadhesive over just 4 weeks. There is still a lack of evidence concerning its long-term risks in local and systemic homeostasis, which should be comprehensively evaluated to promote its clinical translation. Third, the concrete dosage and procedure of OSSA bioadhesive administration and personalized treatment also require careful investigation before clinical translation.

In summary, we reported an in situ injectable OSSA hydrogel bioadhesive based on FDA-approved components of octa-PEG-NH$_2$ and octa-PEG-SSA for the sutureless repair of large gastric defects. Such amide-linked hydrogel was capable of ultrafast in situ gelation, instant adhesion, fluid-tight sealing, and acid-tolerance of gastric perforation, enabling robust wet adhesion interfaces under the fluidically, chemically and mechanically dynamic gastric environments. After a comprehensive evaluation from in vivo and in vitro studies, the OSSA bioadhesive was favorably applied for sutureless repair of gastric perforation in a rat-injured model with neglectable postoperative adhesion, inhibited long-term inflammatory response and interference-free gastric transcriptome and microbiome. Notably, taking advantage of the full set of functionalities, the OSSA bioadhesives could be laparoscopically delivered and robustly adhered to the target defect sites in the porcine stomach with the largest wound size

reported so far, enabling eminent sutureless sealing and efficient repair of gastric perforation defects by a surveillance verification of the combined laparoscopic-endoscopic technique. Despite its early developmental stage, this off-the-shelf bioadhesive with FDA-approved components provides a promising platform for atraumatic sutureless repair of large gastric perforation and offers clinical therapeutic opportunities for other digestive injuries and complications via a minimally invasive technique in diverse scenarios.

## Methods

### Ethical statement
All the animal studies complied with all relevant ethical regulations and were approved by the Ethics Committee of the Animal Centre of Chinese PLA General Hospital (no. 2023-X19-22).

### Materials
Octa-arm poly(ethylene glycol) succinimidyl succinamide (octa-PEG-SSA, Mw = 10 kDa, Mw/Mn = 1.03) and octa-arm PEG-amine (octa-PEG-NH$_2$, Mw = 10 kDa, Mw/Mn = 1.03) were purchased from SINOPEG, China. All chemicals and reagents utilized in the following procedures were analytical grade and used without further purification or treatment.

### Preparation of OSSA hydrogel
Octa-PEG-NH$_2$ and octa-PEG-SSA polymers were dissolved in PBS (pH 7.4) in two sample bottles with the concentration to be 15 wt%, respectively. Then, OSSA hydrogel was obtained by mixing the two components using a dual syringe. When the solutions were no longer flowed backward, the time at which this happened was designated as the gelation time.

### Rheology
The viscoelastic property of OSSA hydrogels was measured using a Thermo Fisher HAAKE MARS III rheometer equipped with parallel plate geometry (20 mm of diameter) at a gap of 2.0 mm. The hydrogels were measured at 25 °C in a frequency range of 1–100 rad s$^{-1}$ at a strain of 1%.

### Compressive behavior
For the compression tests, the OSSA hydrogels were prepared in cylindroid molds (15 mm of diameter and 7.5 mm of height). The compression tests were performed using the universal tensile machine (Instron 3365, USA). A digital caliper was used to measure the dimensions of hydrogels before testing.

### Adhesion assessment
A torsion experiment was conducted to assess the adhesion ability of the OSSA hydrogel to the porcine skin. In brief, the OSSA hydrogel labeled with red dyes was injected onto the tissue surface with a dual syringe. The adhesive flexibility was characterized by imposing the stress of stretching, twisting, and bending onto the tissues. Meanwhile, the OSSA adhesive was exposed to running water to test whether it could be washed away from the adhered porcine skin.

Furthermore, the adhesive property of OSSA hydrogel to porcine tissues was further evaluated by lap shear tests. Generally, fresh porcine skin tissues with rectangle sizes of 60 mm × 20 mm were firstly prepared, and the 50 μL of precursor solutions were applied to two different tissues at one end evenly, followed by attaching together with an adhesive area of 20 mm × 20 mm. The shear strength was then measured using an Instron universal testing machine (Instron 3365, USA) by tensile loading at a strain rate of 1.0 mm/min until detachment was observed. The commercially available tissue adhesives (Coseal, fibrin glue, and Histoacryl) were used as controls. The application of commercially available tissue adhesives followed the provided user guide or operation manuals for each product (measured after 3 min for Coseal, 3 min for fibrin glue, and 1 min for Histoacryl)[29].

## Burst pressure measurement

A section of fresh and cleaned porcine colon was fixed onto the device. A hole of 4 mm in diameter was made with a scalpel and then sealed by 200 μL of OSSA hydrogel in situ to form a thin sealing layer with a thickness of approximately 1 mm. After sealing for 3 min, PBS was pumped into the devices with a syringe pump at a flow velocity of 2 mL/min, and the pressure increased gradually as measured by a connected manometer. The burst pressure was recorded as the maximum pressure before leakage occurred. The commercially available tissue adhesives (Coseal, fibrin glue, and Histoacryl) were used as controls.

## Ex vivo sealing of the gastric perforation

To demonstrate the robust sutureless sealing of gastric perforation by the OSSA bioadhesive, porcine stomachs were purchased and used from local butcher's shop. An incision with a length of 6 cm was created with a scalpel, immediately prior to sealing with an injectable OSSA bioadhesive. Afterwards, the sealed stomach was immersed into PBS buffer, and the stomach cavity was filled with red dye-stained medium of pH 2.0 for better visual to simulate the gastric environment in the body. Images of the sealed incision were taken to monitor the stability of interfacial adhesion for various times. Gastric incision sealed with commercial Coseal was also tested as a control.

## Computational details

Based on the results obtained from experiments, we constructed simplified molecular structural models to analyze the non-covalent bond interactions within the crosslinked networks. Subsequently, geometries of two networks were fully optimized using the B3LYP exchange-correlation functional with Grimme's DFT-D3 (BJ) empirical dispersion correction[51–53], abbreviated as B3LYP-D3 (BJ), and the basis sets was set to 6−31 G(d)[54]. For single-point energy calculations, the more accurate and organics-suited M06-2X functional combined with the def2-TZVP basis set was employed[55,56]. In this work, all DFT calculations were performed using Gaussian 16, Revision A.03[57]. The nature and strength of hydrogen bond interactions was examined by QTAIM[58]. Within QTAIM, the electron density, as well as other real-space functions, such as energy density and the Laplacian of electron density at the so-called BCP of the interatomic interaction of interest, can be comprehensively analyzed. Specific details about QTAIM can be found in the original literature. The analysis of hydrogen bonding and non-covalent interactions was performed using Multiwfn[59].

## Swelling behavior

To characterize the swelling ratio, the initial wet weight of cylindrical OSSA hydrogel (10 mm of diameter and 5 mm of height) was firstly measured ($W_0$). Then, the hydrogels were incubated in 10 mL of pH 7.4 PBS solution, Porcine gastric fluid and human gastric fluid at 37 °C, respectively ($n = 5$). At predetermined times, the weights of the swollen hydrogels were measured ($W_s$) after the surface water was gently removed. The swelling ratio (SR) was calculated as follows:

$$SR(\%) = (W_s - W_0)/W_0 \times 100 \qquad (1)$$

## In vitro biocompatibility

Human gastric mucosa epithelial cell line (GES-1) was employed as the in vitro experimental cells, which were purchased from American Type Culture Collection (CL1317, ATCC, USA). The GES-1 cell line was authenticated with the short tandem repeat method. Dulbecco's modified Eagle medium (DMEM, Gibco, USA) supplemented with 10% fetal bovine serum (Gibco, USA) and 1% penicillin-streptomycin solution (Gibco, USA) was prepared as the complete medium. Cells were grown in the incubator at 37 °C and 5% CO$_2$ atmosphere.

Live/dead, CCK-8 and cell apoptosis assays were conducted to determine the biocompatibility of OSSA hydrogel extracts. GES-1 cells were pretreated with DMEM, Coseal and OSSA hydrogel extracts for 24 h. For the Live/Dead assay, cells were harvested and seeded on 96-well plates and marked according to the manufacturer's instruction (ThermoFisher, USA). The live and dead cells were counted under the fluorescent microscope (Nikon, Japan). For the CCK-8 assay, $2 \times 10^3$ GES-1 cells were seeded in 96-well plates. Cells were incubated with 10% CCK-8 solution (Biorigin, Beijing, China) diluted by DMEM protected from light for 1.5 h. The absorbance of 450 nm was measured by a microplate reader (BioTek, USA). The pretreated cells were collected and stained by AnnexinV-FITC/PI cell apoptosis kit (BD, USA), and the apoptotic rates were determined by a flow cytometer (BD, USA). The gating strategy was determined based on clusters of live and dead cells.

## In vitro and in vivo degradability

To evaluate the in vitro degradation behaviors of the hydrogels, the OSSA hydrogels (10 mm of diameter and 5 mm of height) were firstly lyophilized, and the dried hydrogel was measured as $W_d$. Then, the hydrogels were immersed in pH 7.4 PBS solution, Porcine gastric fluid and human gastric fluid and kept in a shaker incubator at 37 °C. The samples were taken out at specific intervals and weighted ($W_t$) after lyophilization. The mass remaining (%) was calculated as follows:

$$Mass(\%) = W_t/W_d \times 100\% \qquad (2)$$

To investigate the in vivo biodegradability, the anesthetized rats underwent an abdominal incision. The omentum majus was exposed and placed onto a surgical drape. Cy5.5-labeled OSSA hydrogel prepared from octa-PEG-SSA and Cy5.5-modified octa-PEG-NH$_2$ polymers was applied to be injected on the omentum majus by a dual-chamber syringe. After the hydrogel formed robust adhesion and efficient sealing of the wound for 1 min, the omentum majus was put back to its original place and the abdominal incision was sutured. The in vivo degradation behaviors of OSSA hydrogels were characterized by monitoring the fluorescence intensity at a predetermined interval using the in vivo imaging system.

## Animal experiment

Male Sprague-Dawley (SD) rats (250–270 g, 8 weeks) were purchased and housed in the specific pathogen-free condition of the Animal Center of Chinese PLA General Hospital. Bama miniature pigs (20–22 kg, 10 months) were fed in the clean condition. The environment kept 18–22 °C, 40–70% relative humidity and 12/12 h light period. They were given with the sterile feed and water ad arbitrium. The rats and pigs were acclimatized to the feeding condition without any experimental intervention for 1 week to avoid the unpredictable influencing factors before purchase. To avoid the selection bias of in vivo experiments, a random number generation method was adopted, and animals were randomly divided into different groups. The injury modeling, sample collection and determination of experimental indicator levels were blinded with respect to the experimental groups. After the research endpoints, the euthanasia was performed by injection of overdose of pentobarbital sodium.

## In vivo efficacy of the OSSA bioadhesive for sealing gastric defects in rat models

For the preparation of the rat modeling operation, SD rats underwent anesthesia by intraperitoneal injection of 30 mg/kg pentobarbital sodium. The skin in operative regions was shaved and disinfected. Subsequently, the skin, subcutaneous tissues and muscle were successively incised. The stomach was completely exposed and underwent 8 mm incision in the juxtapyloric anterior wall of stomach greater curvature. OSSA bioadhesive was rapidly injected to repair the defect,

while Coseal sealing and manual sutures were also performed as controls. Then, the injured sites received experimental interventions and placed back into the normal anatomical positions. The abdominal wall was closed by manual sutures.

### Laparoscopic delivery of OSSA precursor for sealing gastric defects in porcine models

Pigs received modeling and treatment via laparoscopy. Briefly, two 1-cm holes were generated in the abdomen of anesthetized pigs for constructing laparoscopic routes. After preparation of pneumoperitoneum, the operating arm stuck into the abdominal cavity and 3 cm incision in greater gastric curvature was conducted by electrosurgical knife. Octa-PEG-SSA and octa-PEG-NH$_2$ solutions were simultaneously injected on the injured sites, which were subsequently sealed by OSSA bioadhesive. After the modeling was completed, the holes were stitched, and the pigs were investigated for follow-up. Suture according to the manufacturer's instructions and clinical consensus.

### Repair efficacy via the combined laparoscopic-endoscopic surveillance

Laparoscopic-endoscopic surveillance was conducted to both internally and externally monitor the sealing effects of OSSA bioadhesive for porcine models with gastric defects. Briefly, pigs underwent an incision of approximately in the lower abdomen after general anesthesia. Ten millimeters laparoscopic trocar was inserted via the incision site. Pneumoperitoneum was next established and 4 K high-definition laparoscopic lens was inserted through the trocar for evaluating the repair effects of the gastric serosal surface. Concurrently, a gastrointestinal endoscope lens was inserted through the oral-esophageal route to reach the gastric defect site, monitoring the repair effects of the gastric mucosal surface.

### Histology and immunofluorescence analysis

To investigate the histological changes in tissues, histology and in situ expression of inflammation and fibrosis indicators, fixed tissues with 4% paraformaldehyde were sectioned, deparaffinized and rehydrated by deionized water. H&E staining and Masson staining kits (Solarbio, China) were employed according to the manufacturer's instructions. Histological assessment was independently conducted by two blinded pathologists, and the average scores were used as the final judgment for the staining results. For the immunofluorescence, the rehydrated slides underwent antigen retrieval and 3-cycle washing with PBS-Tween 20 for 5 min. The primary antibodies were used to incubate slides at 4 °C overnight. After repeated a 3-cycle washing, slides were next incubated with secondary antibodies at room temperature protected from light for 2 h. The superfluous reagents were washed out by PBS-Tween 20 for 20 min for three times. Cell nuclei were marked with DAPI (Solarbio, China). The fluorescent images were photographed under a laser confocal microscope (Nikon, Japan). The fluorescent intensities were quantified using ImageJ software. Specifically, the integrated densities were determined after ascertaining the thresholds of positive fluorescent signals after the images were transformed to the 8-bit type. The relative fluorescent intensities were normalized for the statistical comparisons among groups.

### Detection of blood sample

To investigate the effects of gastric fistula and materials on rats, the complete blood routine counts, biochemical indexes, and inflammatory responses were detected. The whole blood was harvested from the retro-orbital venous plexus of rats. For the conduction of complete blood routine counts and biochemical index tests, the whole blood samples were stored in the sodium citrate anticoagulated tubes, which was then left static for 30 min. The samples were detected by TEK-II full-automatic analyzer (Tekang, Jiangxi, China) for routine blood test and an automatic biochemical analyzer (Hitachi, Japan).

We chose plasma TNF-α and IL-6 cytokines as indicators of inflammatory responses. The whole blood of the rat was centrifuged at 1200 × $g$ for 10 min at 4 °C, and the supernatant was harvested. It was then centrifuged again to obtain plasma samples. TNF-α and IL-6 were detected according to the manufacture's protocols (Abcam, UK).

### Transcriptome analysis of gastric tissues

To investigate the impact of surgical suture, Coseal and OSSA on stomach tissues, RNA-seq was performed for depicting landscapes of transcriptome of the experimental groups. Briefly, total RNA was extracted with Trizol agent. The purities and concentrations were determined using Agilent4200 (Agilent, Germany) and Qubit (ThermoFisher, USA). The poly-A tail was bound with magnetic beads with oligo dT for the purification of mRNA. The harvested mRNA was randomly segmented and reversely transcribed into cDNA. The cDNA libraries received joint connection with the insertion of index primers and sequenced with the Illumina Novaseq 6000 platform (CA, USA). The transcriptome data, revealing differentially expressed genes, were used to show the effects of the repairing methods.

### Investigations into landscapes of gastric bacterial communities

The alterations in microbiome likewise served as an index to evaluate the safety of repairing materials. Total DNA was extracted using genomic DNA extraction kit (Qiangen, Germany). The DNA served as the templates for the amplification of 16S rRNA genes. The magnetic beads were used to purify and recycle the amplification products, which was then quantified with Quant-iT PicoGreen dsDNA Assay Kit (Invitrogen, USA) and a microplate reader (BioTek, USA). TruSeq Nano DNA LT Library Prep Kit (Illumina, USA) was used to construct cDNA libraries. 16s rRNA sequencing was conducted with the Illumina Novaseq platform (USA).

### Statistical analysis

Data with normal distributions were presented as mean ± standard error of mean (SEM). The statistical differences between the two groups were analyzed using two-sided Student's $t$ test. Comparing differences among multiple differences was performed using one-way analysis of variance (ANOVA) followed by the least significant difference post hoc test. GraphPad Prism 8.0 and SPSS 25.0 were employed to assess the statistical significance. $P < 0.05$ was regarded as the threshold of significant difference. The pre-test studies were conducted for sample size calculation. A hazard ratio of 0.15, 80% power and two-sided alpha of 0.05 served as basic parameters and the sample sizes were calculated using PASS 15.0 (NCSS Corp.).

### Reporting summary

Further information on research design is available in the Nature Portfolio Reporting Summary linked to this article.

## Data availability

The authors declare that all data needed to support the findings of this study are available within the article, Supplementary information, and Source data file. RNA-Seq and 16s rRNA-seq data generated have been deposited in the NCBI SRA database under the accession code PRJNA1426413. Any additional information required to reanalyze the data reported in this work is available from the corresponding authors upon request. Source data are provided with this paper.

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

## Acknowledgements

This work was supported by the National Natural Science Foundation of China (no. 52573186, X.W.; 52373162, X.W.; 82572862, B.W.), Beijing Natural Science Foundation (no. F252056, X.W.; L256034, J.C.; L244037, X.W.; 7262018, B.C.) and Beijing Nova Program (no. 2024048519, L.L.).

## Author contributions

Z.W., J.C., and X.W. designed the whole work and wrote the manuscript. Z.W., B.C., and L.L. carried out the experiments and analyzed data. H.C., B.W. performed the characterization and analyzed data. All authors commented on the manuscript.

## Competing interests

The authors declare no competing interests.

## Additional information

**Supplementary information** The online version contains Supplementary material available at https://doi.org/10.1038/s41467-026-71031-9.

