## [Transparent Peer Review File · Nature Communications]

Acid-tolerant injectable bioadhesive for sutureless repair of large gastric perforation

Corresponding Author: Professor Xing Wang

Version 0:

Reviewer comments:

Reviewer #1

(Remarks to the Author)

Wang et al reported a kind of injectable acid-tolerant hydrogels for sutureless repair of large gastric defects, by using the octa-armed poly(ethylene glycol) succinimidyl succinamide (Octa-PEG-SSA) and octa-armed poly(ethylene glycol) amine (Octa-PEG-NH₂). The hydrogel bioadhesives exhibited injectable shape adaptability, ultrafast gelation, good biocompatibility and instant wet adhesion, and also stability against gastric fluid, allowing for the use for gastric defects. However, the level of advancement of this work can not justify its publication in nature communications, since

1. hydrogel bioadhesives have been widely used for gastric defects. Even though the authors have claimed the targeted large-size gastric defect, the 8-mm incision did not represent a great technical advancement, since 5-mm incision model has been used in many previous reports, for example Science Translational Medicine 14, no. 630 (2022): eabh2857.
2. the authors have previously reported the used of tetra-PEG based hydrogel bioadhesives for tissue adhesion and also gastric defect sealing. The change from tetra-PEG from Octa-PEG does not carry the significant advancement.
3. all those physicochemical, mechanical, and adhesive performances studies were similar to those other hydrogel bioadhesives systems for gastric sealing. The authors should justify why previous hydrogel bioadhesives failed for large-size gastric defect sealing, and which unique characters of their hydrogel bioadhesives could overcome such challenge and enable the successful sealing. And also why octa-armed PEG hydrogels deliver such unique performances.

Reviewer #2

(Remarks to the Author)

This manuscript designs a novel injectable OSSA (Octa-PEG-SSA and Octa-PEG-NH₂) hydrogel bioadhesive for the sutureless repair of large gastric defects. The hydrogel bioadhesive demonstrates rapid in situ gelation, strong tissue adhesion under extreme acidic conditions, minimal swelling, and biocompatible degradation. The efficacy is validated across ex vivo and in vivo (rat and porcine) models using mechanical, histological, immunological, transcriptomic, and microbiome evaluations. This manuscript presents a comprehensive and promising contribution to bioadhesive development for gastrointestinal surgery. Addressing the below concerns will significantly strengthen its impact and translational value. The comments for this manuscript are listed as follows:

1. Authors need to compare the adhesion properties of OSSA hydrogel and commercial products.
2. In in vitro and vivo experiments, the authors have shown that hydrogel can strongly bond and shown wound sealing property. Can this injectable bioadhesive glue be leakage from defect area? what is probability of postoperative adhesion? Some experiments that can show the comparative advantages of this injectable glue and adhesive tape are needed.
3. The author needs to provide detailed information on the burst pressure test parameters. For example, the amount of hydrogel applied, the thickness of the cured glue, etc.
4. In Fig.5, In vitro and in vivo biodegradation and biocompatibility of the OSSA hydrogel, the image of cell viability in Fig 5f shows that Coseal group has much more alive cells than control group while the quantity analysis did not have the consistent result. Additionally, the author claim it has long term biocompatibility, while cell viability and toxicity only show 24h data. Besides these, statistical methods (e.g., sample size calculation, blinding) are not described in sufficient detail in vivo studies. Also, immunohistochemistry analysis should be added to check the immune responses after applying the glue and compare it with commercial products.
5. In Fig.6, In vivo adhesion ability of the OSSA bioadhesive in rat models, the low inflammatory response of the glue is reported based on histology images, but it is not clear to see the injured sites of stomach from HE and IHF images, author

should mark the injured tissues healed area and residual materials in the Fig6 c, and the IHF images are not informative, should provide additional large magnificent images for injured tissue healing area. Besides these, authors should add quantitative analysis of IHF data, and also need clearly define statistical analysis methods.

6. For OSSA bioadhesive in sealing large gastric defects in porcine models in Fig7f, it is also difficult to find the injured sites of stomach and residual materials for OSSA group in HE and Masson images, the images show the healthy stomach tissues. Control group (commercial products) should be added in Figure 7 for the in vivo tests. Also, immunohistochemistry analysis should be added to check the immune responses after applying the hydrogel glue and compare it with commercial products.

Version 1:

Reviewer comments:

Reviewer #1

(Remarks to the Author)

While they emphasize the advantage of the hydrogel adhesives, (i.e., operation convenience, instant wet adhesion, extreme acid/pepsin resistance, in vitro biocompatibility, biodegradability, inflammatory response, postoperative adhesion prevention, in vivo therapeutic efficiency in larger gastric defects, and product stability), these contributions appear insufficient to meet the rigorous novelty standards typically required for publication in Nature Communications. The reported advancements seem incremental rather than transformative, falling short of the groundbreaking innovations expected in such a high-impact journal.

Even though they claim that “it is difficult to persistently adhere to the harsh gastric defects under strong acid, digestive enzymes, and mechanically dynamic environments for gastric repair”, as I mentioned before, similar works have been widely reported for such application. The change from tetra-PEG to octa-PEG hydrogel is a kind of simply extension from one material system to another. Increase in gastric defect can not represent a key advancement in the scientific contribution. Some underlying mechanism why such kind of change can improve the performance has not been clearly clarified.

Reviewer #2

(Remarks to the Author)

The manuscript has improved substantially after the authors addressed the reviewers' comments. However, the schematic figures appear to have been generated using AI tools, and the source or method of generation has not been disclosed. The authors should provide clarification and include appropriate attribution in accordance with the journal's guidelines.

Version 2:

Reviewer comments:

Reviewer #1

(Remarks to the Author)

The authors have addressed some of the initial queries, but several critical issues remain unresolved, preventing me from recommending the article for publication. Regarding their claim that current hydrogel sealants, primarily based on hydrolytically labile bonds, cannot reliably endure stability in extremely acidic fluids and digestive enzymes, this assertion is contradicted by a substantial body of literatures. Numerous studies, such as Science Translational Medicine 12, 558 (2020): eaba8014; Science advances 7, no. 23 (2021): eabe8739; Science translational medicine 14, 630 (2022): eabh2857; ACS nano 17, 1 (2022): 111-126; Advanced Functional Materials 32, 29 (2022): 2202285; Advanced Functional Materials 34, 48 (2024): 2408479; Chinese Journal of Chemistry, 41, 23 (2023): 3339-3348; ACS Nano 17, 10 (2023): 9521-9528; Science Translational Medicine 17, 787 (2025): eadq1975; to cite only a few, demonstrate the successful use of hydrogel bioadhesives with long-term resistance to gastric fluids. A comprehensive review in Theranostics.16(2026):2221–2248 further supports this. Thus, the authors' claim that most hydrogels fail in such environments is unfounded, as viable alternatives are well-documented.

The authors also argue that adhesives capable of maintaining structural stability in the harsh gastric environment are either difficult to degrade or produce toxic degradation products. However, this generalization is problematic. Many of the cited successful systems neither face these challenges nor exhibit such limitations. The authors should re-examine these examples to avoid making sweeping and unsubstantiated claims about the degradability and toxicity of existing hydrogels.

While the authors highlight that their study involves the largest defect ever reported to be sealed by a hydrogel bioadhesive in a large animal model, this alone does not justify groundbreaking innovation. The increase in defect size does not inherently demonstrate why previous systems would fail at this scale or whether the limitation was merely due to the absence of such large models in prior studies. Without this justification, the novelty and impact of their work remain unclear.

After a thorough review of the authors' response and the revised manuscript, I conclude that the study lacks sufficient strengths to meet the high standards of the journal. I regret to say that, the unresolved issues and unsubstantiated claims undermine the manuscript's potential for acceptance.

A point-by-point response to the reviewer

Dear the Reviewers:

Thank you very much for your kind advice and valuable comments in helping us improve our manuscript (Manuscript ID: NCOMMS-25-23781A-Z; Title: “In situ injectable bioadhesive with instant wet adhesion and extreme acid-tolerance for sutureless repair of large gastric perforation”). We have substantially modified the manuscript, according to the questions raised by the Reviewers. **All the modified words, sentences and paragraphs were labeled with red fonts.** A point-to-point response to highlight how we have addressed each of the comments is listed below.

Reviewers' comments:

Reviewer #1 (Remarks to the Author):

Wang et al reported a kind of injectable acid-tolerant hydrogels for sutureless repair of large gastric defects, by using the octa-armed poly(ethylene glycol) succinimidyl succinamide (Octa-PEG-SSA) and octa-armed poly(ethylene glycol) amine (Octa-PEG-NH₂). The hydrogel bioadhesives exhibited injectable shape adaptability, ultrafast gelation, good biocompatibility and instant wet adhesion, and also stability against gastric fluid, allowing for the use for gastric defects. However, the level of advancement of this work can not justify its publication in nature communications, since

1. hydrogel bioadhesives have been widely used for gastric defects. Even though the authors have claimed the targeted large-size gastric defect, the 8-mm incision did not represent a great technical advancement, since 5-mm incision model has been used in many previous reports, for example Science Translational Medicine 14, no. 630 (2022): eabh2857.

Reply 1: Due to the small volume of the rat stomach, we could only make an 8-mm incision on the rat models. However, the OSSA bioadhesive was capable of sealing injury of *in vivo* porcine gastric defects of 3 cm in our manuscript, far exceeding the currently reported gastric defect models including the Sci. Transl. Med. 2022, 14, eabh2857. Furthermore, to the best of our knowledge, the largest length of wound for porcine gastric defect modeling is 1 cm (ACS Nano 2025, 19, 17533; ACS Nano 2023, 17, 111). Until now, there is a lack of research on testing the sealing effect of wounds larger than 1 cm using porcine models. So, we used ‘large-size’ to describe to sealing advantages of the OSSA hydrogel.

2. the authors have previously reported the used of tetra-PEG based hydrogel bioadhesives for tissue adhesion and also gastric defect sealing. The change from tetra-PEG from Octa-PEG does not carry the significant advancement.

Reply 2: Thanks for your kind suggestions. Although the previously reported tetra-PEG based hydrogel bioadhesives were used for the tissue defect sealing, these features only meet the basic properties required for medical sealants. However, it is difficult to persistently adhere to the harsh gastric defects under strong acid, digestive enzymes, and mechanically dynamic environments for gastric repair. So, it is not a simple adjustment from tetra-PEG to octa-PEG hydrogel. On the contrary, unlike the easily degradable ester bonds by acidic or enzymatic hydrolysis for previously reported tetra-PEG hydrogels, the stable amide-linked skeleton and strong hydrogen bonding interactions within the dense crosslinking network furnished the OSSA hydrogel with excellent stability against gastric fluid/pepsin, low swelling ratio and slow degradable deterioration in strength and interfacial robustness, thus accommodating the long-term adhesion on the gastric tissues under wet and dynamic motion of the porcine stomach.

3. all those physicochemical, mechanical, and adhesive performances studies were similar to those other hydrogel bioadhesives systems for gastric sealing. The authors should justify why previous hydrogel bioadhesives failed for large-size gastric defect sealing, and which unique characters of their hydrogel bioadhesives could overcome such challenge and enable the successful sealing. And also why octa-armed PEG hydrogels deliver such unique performances.

Reply 3: Thanks for your kind suggestions. As mentioned above, gastric juice contained a lot of extremely acidic fluids, digestive enzymes, intrinsic factors and mucoproteins that could rapidly destroy the majority of natural polysaccharide linked by glucoside bonds and synthetic degradable polymers linked by ester bonds through acidic or enzymatic hydrolysis, thus accelerating the degradability of those previous hydrogel bioadhesives and impairing their sealing effects before the gastric healing, which may lead to the probable leakage of gastric contents and peritonitis from larger diameter of gastric defects. In contrast, the extreme acid/pepsin-tolerance of OSSA hydrogel was mainly attributed to its stable multi-amide linkages and regionally concentrated hydrogen bonding interactions throughout the dense network, which could significantly hinder the fluid penetration and other soluble substances. Therefore, the injectable OSSA bioadhesive can be laparoscopically delivered and robustly adhered to the target defect sites in the porcine stomach with the largest size of gastric defects

so far (3 cm), enabling eminent sutureless sealing and efficient repair of gastric defects.

More importantly, after comprehensive comparison and evaluation of the OSSA bioadhesive with the commercially available tissue adhesives, OSSA bioadhesive could significantly outperform the clinic surgical sutures and existing tissue sealants in terms of operation convenience, instant wet adhesion, extreme acid/pepsin resistance, in vitro biocompatibility, biodegradability, inflammatory response, postoperative adhesion prevention, in vivo therapeutic efficiency in larger gastric defects, and product stability. Collectively, it can be reasonably concluded that this high-performance bioadhesive with FDA-approved PEG components will provide a clinically feasible and effective therapeutic platform for the atraumatic sutureless repair of large-size gastric perforation, and offer clinical therapeutic opportunities for other digestive injuries and/or complications through minimally invasive techniques in diverse scenarios.

Reviewer #2 (Remarks to the Author):

This manuscript designs a novel injectable OSSA (Octa-PEG-SSA and Octa-PEG-NH₂) hydrogel bioadhesive for the sutureless repair of large gastric defects. The hydrogel bioadhesive demonstrates rapid in situ gelation, strong tissue adhesion under extreme acidic conditions, minimal swelling, and biocompatible degradation. The efficacy is validated across ex vivo and in vivo (rat and porcine) models using mechanical, histological, immunological, transcriptomic, and microbiome evaluations. This manuscript presents a comprehensive and promising contribution to bioadhesive development for gastrointestinal surgery. Addressing the below concerns will significantly strengthen its impact and translational value. The comments for this manuscript are listed as follows:

1. Authors need to compare the adhesion properties of OSSA hydrogel and commercial products.

Reply 1: Thanks for your kind suggestions. We have compared the adhesion properties of OSSA hydrogel and commercial products accordingly in the revised manuscript. The shear adhesive strength of OSSA hydrogel could achieve 34.0 ± 2.3 kPa, superior to the commercially available tissue adhesives and sealants, including the Coseal (10.3 ± 0.9 kPa), Fibrin glue (10.4 ± 0.6 kPa), and cyanoacrylate glue (Histoacryl, 23.6 ± 1.5 kPa). Similarly, the interfacial toughness was more than 118.2 ± 10.2 J m⁻² in comparison with those of Coseal (23.1 ± 3.4 J m⁻²), Fibrin glue (24.8 ± 2.9 J m⁻²), and Histoacryl (39.7 ± 3.3 J m⁻²). In addition, the burst pressure of OSSA hydrogel reached

19.2 ± 2.36 kPa, which also far exceed those of Coseal (6.12 ± 1.01 kPa), Fibrin glue (5.64 ± 0.36 kPa), Histoacryl (6.88 ± 0.88 kPa), and the *in-vivo* gastrointestinal pressure (0.5~4.0 kPa) of normal human.

2. In *in vitro* and *in vivo* experiments, the authors have shown that hydrogel can strongly bond and shown wound sealing property. Can this injectable bioadhesive glue be leakage from defect area? what is probability of postoperative adhesion? Some experiments that can show the comparative advantages of this injectable glue and adhesive tape are needed.

Reply 2: Thanks for your kind suggestions. To address the concern of potential hydrogel leakage, we conducted a specific leakage assessment using *ex vivo* porcine gastric defect models with larger incisions (~20 mm) to simulate the clinically relevant full-thickness perforations. As clearly demonstrated in the revised Supplementary Movie 4, the injectable OSSA bioadhesive labeled with blue dye was rapidly deposited *in situ* and robustly adhered onto the damaged tissue, and no significant glue leakage or other gel-like substance was observed from the defect regions, corroborating its instant adhesion, fluid-tight capacity, and suitability for sutureless sealing of large gastric perforation.

To assess the risk of postoperative adhesion, we conducted the blinded laparoscopic evaluations in porcine model using the validated modified American Fertility Society (mAFS) scoring system, as shown in the revised Supplementary Fig. 21. Two independent investigators quantitatively assessed the postoperative adhesion through the intensity, scope, and affected area metrics. It was noteworthy that OSSA glue not only achieved superior repair efficacy comparable to the clinical suture method in terms of gastric defect repair, but also had significantly lower levels of abdominal adhesions. Compared to the control groups with varying degrees of severe adhesions and organ displacement, the OSSA group had the lowest score for abdominal adhesions. This remarkable anti-adhesion performance had a dual function on effectively preventing pathological adhesions at the repair sites and preserving the integrity of the surrounding tissues, which offered a significant and innovate advantage over existing solutions. Therefore, this double-sided or Janus-like OSSA bioadhesive had significant potential in gastrointestinal repair and postoperative adhesion prevention.

As for the comparison with the adhesive tape, we have summarized the following several important differences although no extensive experiments have been conducted for verification: 1) Usage convenience and shape adaptation. The current adhesive tape

is generally difficult to be applied to more geometrically and anatomically complex defects while the OSSA glue can be feasibly injected to fill the irregular defects before the gelation, thereby allowing the consequent adaptation into various defected regions with geometrically and anatomically complex shapes; 2) Effectiveness and stability of gastric defect sealing in wet environments. Compared to the adhesive tapes with the physical, slow, and insecure tissue adhesion, the OSSA glue without relying on the external devices and complicated preparation methods (e.g., UV source, crosslinking agents or initiators, and pressurized air-based blow spinner) exhibits robust wet tissue adhesion and instant closure of the irregular gastric defects due to its subsequent procedures of rapid absorption and drying of interfacial water, effective chemical reaction, and rapid anchoring onto the wet gastric tissues upon contact, which can prevent the bleeding and leakage of bodily fluids more stably and persistently, especially in harsh gastric conditions or other wet and dynamic biological environments; 3) Delivery convenience and self-removability *in vivo*. The OSSA glue can be facilely delivered via a minimally invasive technique *in vivo* and robustly adhered onto the slippery gastric defects within seconds, but the adhesive tapes may require complex, time-consuming and or invasive sealing procedures. In addition, self-biodegradable feature *in situ* and nontoxic degradable products of OSSA can efficiently prevent infections caused by the removal of additional materials (e.g., duct tapes) during a secondary surgery, thereby offering a facile, atraumatic, safe, fluid-tight, robust, and sutureless sealing of large gastric defects.

It is noteworthy that an innovate off-the-shelf gastrointestinal patch proposed by Prof. Xuanhe Zhao exhibited excellent performance and remarkable sutureless repair of gastrointestinal defects with significantly clinical applications (Sci. Transl. Med. 2022, 14, eabh2857). Inspired by its convenience and effectiveness of GI patch in nonmedical applications, our designed OSSA glue is also expected to evolve into another promising alternative to suture for the repair of large gastric defects and offer potential clinical opportunities for the repair of other organs and injuries in the human body.

3. The author needs to provide detailed information on the burst pressure test parameters. For example, the amount of hydrogel applied, the thickness of the cured glue, etc.

Reply 3: Thanks for your kind suggestions. A hole of 4 mm in diameter was made on a porcine colon with a scalpel and then sealed by 200 μ L of OSSA hydrogel *in situ* to form a thin sealing layer (thickness: \sim 1 mm). After sealing for 3 min, PBS was pumped into the devices with a syringe pump at a flow velocity of 2 mL/min and the pressure

increased gradually as measured by a connected manometer. In fact, the amount of the injectable hydrogel glue can be flexibly adjusted according to the size and shape of the defects. Meanwhile, since the glue may be applied around the damaged area during the injecting process, the thickness and size of the glue layer can also be locally trimmed or partially removed according to the actual situation, thus providing more convenience for clinical use.

4. In Fig.5, In vitro and in vivo biodegradation and biocompatibility of the OSSA hydrogel, the image of cell viability in Fig 5f shows that Coseal group has much more alive cells than control group while the quantity analysis did not have the consistent result. Additionally, the author claim it has long term biocompatibility, while cell viability and toxicity only show 24h data. Besides these, statistical methods (e.g., sample size calculation, blinding) are not described in sufficient detail in vivo studies. Also, immunohistochemistry analysis should be added to check the immune responses after applying the glue and compare it with commercial products.

Reply 4: We sincerely thank the reviewer for your thoughtful observation regarding the apparent discrepancy between live/dead imaging and quantitative viability. The differences in cell densities in the representative image are mainly due to the unevenness of cell adhesion. In fact, comparison of the total amount of cells has limited significance and the ratio of dead and live cells is the key indicator. It can reflect the potential effects of bioadhesives on cell viability and proliferation, and is not affected by the differences in total number of seeded cells. Even so, we have further expanded the scope of biocompatibility assessment for 72 h through comprehensive Live/Dead imaging, CCK-8 assays and apoptosis analysis using human gastric mucosa epithelial cells (GES-1). Compared to the Coseal group, the OSSA group exhibited higher cell viability and lower cell apoptosis of GES-1, analogous to the control group, indicating the insignificant impact of OSSA on the maintenance and proliferation gastric mucosa epithelial cells. Additionally, no obvious differences in cell densities and morphologies were observed between the OSSA hydrogel and the control groups, as well as almost no dead cells (red fluorescence) were detected, further validating that OSSA hydrogel had excellent biosafety on survival and functions of stomach epithelial cells in vitro.

In addition, we have rigorously implemented and documented enhanced statistical protocols according to your suggestions. Finally, the histological examinations and immunohistochemistry analysis comparing OSSA hydrogel against the commercial Coseal was also performed in the revised Supplementary Fig. 10. These results showed

that OSSA glue and Coseal were comparable in terms of inflammatory manifestations during the initial period, with no significant difference in immune cell activation or cytokine expression, reflecting the favorable immune-compatibility of the OSSA glue.

5. In Fig.6, In vivo adhesion ability of the OSSA bioadhesive in rat models, the low inflammatory response of the glue is reported based on histology images, but it is not clear to see the injured sites of stomach from HE and IHF images, author should mark the injured tissues healed area and residual materials in the Fig6 c, and the IHF images are not informative, should provide additional large magnificent images for injured tissue healing area. Besides these, authors should add quantitative analysis of IHF data, and also need clearly define statistical analysis methods.

Reply 5: Thanks for your kind suggestions. We sincerely apologize for this oversight where specific information was not marked in the histological and immunofluorescent images. The injured sites and residual biomaterials have been added accordingly in the revised manuscript. We simultaneously showed the macroscopic and local images to further display more information of pathological alterations. The large magnificent IHF images, quantitative analysis of IHF data and the statistical analysis methods have also been added according to your valuable suggestions.

6. For OSSA bioadhesive in sealing large gastric defects in porcine models in Fig7f, it is also difficult to find the injured sites of stomach and residual materials for OSSA group in HE and Masson images, the images show the healthy stomach tissues. Control group (commercial products) should be added in Figure 7 for the in vivo tests. Also, immunohistochemistry analysis should be added to check the immune responses after applying the hydrogel glue and compare it with commercial products.

Reply 6: We sincerely thank the reviewer for your thoughtful comments and kind suggestions. The injured sites of stomach and residual materials have been labelled for OSSA group in the revised HE and Masson images. In addition, we have added two commercial products of Coseal and Histoacryl as control adhesives in the evaluation of repair effect using porcine models, in addition to the suture group. The results showed that OSSA glue could significantly outperform the Coseal and Histoacryl adhesives in sealing efficacies, biocompatibility, and long-term postoperative anti-adhesion. Also, we employed the immunofluorescence staining and normalized immunofluorescence intensity analysis to evaluate the immune responses. The results displayed that the macrophage markers of collagen I, T cells (CD3), and α -SMA were all lower than those in the suture, Coseal, and Histoacryl (Fig. 7F and Supplementary Fig. 22), further

indicating a lower degree of fibrosis and inflammatory response in the OSSA group during the long-term healing process, in agreement with the fibrosis and inflammation observed in the histological evaluation.

Collectively, this OSSA bioadhesive embodies multifunctionalities to not only synergistically solve the key limitations of surgical suture and commercially available tissue sealants, but also initiatively achieve instant wet adhesion and durable closure of large-size porcine gastric perforations via laparoscopic delivery and intuitively monitor the long-term sealing efficacy in real time via the combined laparoscopic-endoscopic technique. So, this work will open a feasible and universal avenue to create an in situ injectable glue with robust wet adhesion and extreme acid-tolerance for sutureless repair of large gastric defects and other digestive complications via a minimally invasive technique.

Finally, we sincerely thanks for your thorough review of our manuscript and your valuable suggestions to strengthen the impact and translational value, and hope our explanation can get your understanding and recognition. We recognize that this feedback is of great importance in further refining and enhancing our research outcomes.

A point-by-point response to the reviewer

Dear the Reviewers:

Thank you very much for your kind advice and valuable comments in helping us improve our manuscript (Manuscript ID: NCOMMS-25-23781A-Z; Title: “In situ injectable bioadhesive with instant wet adhesion and extreme acid-tolerance for sutureless repair of large gastric perforation”). We have substantially modified the manuscript, according to the questions raised by the Reviewers. **All the modified words, sentences and paragraphs were labeled with red fonts.** A point-to-point response to highlight how we have addressed each of the comments is listed below.

Reviewers' comments:

Reviewer #1 (Remarks to the Author):

While they emphasize the advantage of the hydrogel adhesives, (i.e., operation convenience, instant wet adhesion, extreme acid/pepsin resistance, in vitro biocompatibility, biodegradability, inflammatory response, postoperative adhesion prevention, in vivo therapeutic efficiency in larger gastric defects, and product stability), these contributions appear insufficient to meet the rigorous novelty standards typically required for publication in Nature Communications. The reported advancements seem incremental rather than transformative, falling short of the groundbreaking innovations expected in such a high-impact journal.

Even though they claim that “it is difficult to persistently adhere to the harsh gastric defects under strong acid, digestive enzymes, and mechanically dynamic environments for gastric repair”, as I mentioned before, similar works have been widely reported for such application. The change from tetra-PEG to octa-PEG hydrogel is a kind of simply extension from one material system to another. Increase in gastric defect can not represent a key advancement in the scientific contribution. Some underlying mechanism why such kind of change can improve the performance has not been clearly clarified.

Reply: We sincerely thank the reviewer for this second round of critical comments and the opportunity to further clarify the transformative nature of our work. We understand the reviewer's emphasis on the high novelty threshold for *Nature Communications*. We respectfully argue that our advancement is not incremental but represents a paradigm shift in the design philosophy and clinical capability of gastric bioadhesives. More specifically the key points of our study are:

1. Addressing a critical clinical unmet need, not just a material variation

Although the previously reported hydrogel bioadhesives (including our own tetra-PEG systems) have shown utility in idealized or small-scale injury scenarios, their susceptibility to rapid degradation with short-term tolerance in harsh gastric fluid environments (acids, enzymes, dynamic motion) has consistently limited their application to low-risk small perforations. The central challenge for large-size defects (>1 cm in large animals) is sustained, failure-proof sealing over the critical healing period to prevent life-threatening peritonitis. Current hydrogel sealant, mainly based on hydrolytically labile bonds (e.g., esters in many synthetic polymers or glycosidic bonds in polysaccharides), cannot reliably endure

stability in extremely acidic fluids and digestive enzymes. While another series of adhesives capable of maintaining structural stability in the harsh gastric environment are difficult to degrade or the degradation products are inevitably toxic, which pose a serious potential risk *in vivo*. Therefore, the aforementioned two types of reported hydrogel bioadhesives and clinical sealants cannot reliably meet this demand.

Of note, we have never reported the tetra-PEG ester network for gastric sealing, and the other reported tetra-PEG hydrogels are also not capable of sealing large size of gastric defects, because these ester-based PEG networks are definitely not qualified for persistently adhering to the target sites under strong acid, digestive enzymes, and dynamic environments. Consequently, compared to the ester-based PEG network, our amide-based PEG network (below Figure 1) is a fundamental redesign for constructing stable networks to directly conquer the clinical challenge of durable stability for a long period (more than 4 weeks), particularly in the irregular surface of gastric tracts and the fluidically, chemically and mechanically dynamic in-vivo environment. Therefore, it is not a simple extrapolation but a deliberate move from a temporarily adhesive scaffold to a persistently robust sealant.

Figure 1. Schematic diagram of intermediate structural variation of the OSS (left) and OSSA (right) hydrogels. Left: crosslinking junction of ester and amide bonds with loose hydrogen bonding domains. Right: crosslinking junction of only amide bonds with dense hydrogen bonding domains.

To further distinguish the differences between ester-based PEG network and amide-based PEG network, we systematically compared our octa-PEG (OSSA) hydrogel with previously reported ester-based octa-PEG (OSS) hydrogel under identical harsh conditions (simulated gastric fluid, pH 1.5 with pepsin). As previously discussed, OSSA hydrogel can maintain a structural stability and durable wet adhesion onto porcine skin with negligible change in mass, size and transparency and maintainable tissue adhesion within 14 days (Supplementary Figs. 5, 6). In contrast, the OSS hydrogel with the same solid contents is completely degraded within 2 days (below Figure 2A) due to the rapid hydrolysis of ester groups in such harsh SGF conditions, demonstrating the interfacial adhesion failure *in vitro* and inferring the impossibility on sealing gastric perforations under the extremely acidic fluids *in vivo* for a long period.

Additionally, the clinical inadequacy of ester-based sealants for gastric applications have actually been evidenced by our direct benchmarking against Coseal, a commercial tetra-PEG sealant. While effective in other surgical contexts, its ester-linked network proves the fundamentally unstable in the large gastric models, failing rapidly due to its acidic hydrolysis (below Figure 2B), which can be verified by its weak adhesion stability, mechanical robustness and sealing capacity from both *in vitro* and *in vivo*

results in this study. This critical failure of a state-of-the-art commercial product highlights a previously unaddressed limitation in the field and defines the specific clinical problem our work aims to solve.

Figure 2. Representative images of (A) OSS hydrogel and (B) Coseal sealant adhering onto the porcine skin immersed in simulated gastric fluids for 2 days.

Consequently, this direct comparison unequivocally proves that OSSA is not "similar" to previous works; it operates on a completely different level of performance and stability, which is the fundamental reason why it succeeds in sealing large defects (3 cm) where others fail. Therefore, the transition to our stable amide-linked OSSA network is not an incremental improvement but a necessary redesign to enable a previously impossible clinical application.

2. Elucidating the underlying mechanism for performance leap

The underlying mechanism of this exceptional performance originates from the synergistic effect of two key design elements: chemically stable backbone and dense hydrogen bonding. On the one hand, replacing easily hydrolyzable ester linkages with extremely stable amide linkages provides intrinsic resistance to acidic and enzymatic hydrolysis. Monitoring the degradation in acid shows that the ester-based OSS hydrogel rapidly breaks down via the ester hydrolysis, thereby releasing some soluble PEG fragments (e. g., octa-PEG-OH, octa-PEG-COOH). The entirely disappeared characteristic peak of CH₂ groups (*a*) next to the oxygen atom of ester groups in ¹H NMR spectra (below Figure 3) clearly pronounces the complete fracture of ester bonds, indicative of the acid-induced degradation of OSS hydrogel. While the OSSA bioadhesives can remain intact with no detectable soluble degradation products over the same period, revealing the chemically stable amide bonds in acidic conditions.

Figure 3. ^1H NMR spectra of Octa-PEG-SS-succinate polymer and degradation products of OSS hydrogel after incubation in acidic conditions for 2 days.

On the other hand, the densely crosslinked network with concentrated hydrogen bonding in octa-armed architecture enables the formation of a much more densely crosslinked network upon gelation. This dense network, fortified by regionally concentrated hydrogen bonding, acts as a formidable barrier against gastric fluid penetration and chain disentanglement, leading to exceptionally lower swelling and slower degradation. Hydrogen bonding interactions significantly affected the infrared characteristic peaks of amide bonds, which are mainly manifested as the red shift and peak shape changes of N-H and C=O stretching vibrations (Protein Sci. 2003, 12, 520; Macromolecules 1985, 18, 1676). FTIR analysis confirms the presence of stable amide linkages and indicates strong hydrogen bonding within the OSSA network. As shown in below Figure 4, the formation of typical new peak ($\nu_{\text{C=O}}$) at 1678 cm^{-1} , widening peak shape ($\nu_{\text{N-H}}$) at $3140\text{-}1740\text{ cm}^{-1}$ and obvious red shift collectively verified the powerfully intermolecular hydrogen bonding interactions among the stable amide-linked skeleton of OSSA hydrogel, compared to the ester-linked OSS hydrogel.

Figure 4. FTIR spectra of OSS and OSSA hydrogels.

To gain atomic-level insight into the crosslinking network, we then construct simplified molecular models to probe the non-covalent interactions within the crosslinked networks at the atomic level. The models were designed to isolate the effect of the crosslinking chemistry: both models featured an identical octa-armed PEG core but were connected via either ester and amide linkages (OSS Network) or only amide linkages (OSSA Network), respectively. The geometries of both networks were fully optimized using the B3LYP-D3 (BJ) and the basis sets was set to 6-31G(d). For more accurate single-point energy calculations, the organics-suited M06-2X function combined with the def2-TZVP basis set was employed. All DFT calculations were performed using Gaussian 16. The nature and strength of hydrogen bond interactions were examined by the quantum theory of atoms-in-molecules (QTAIM) analysis, by which the electron density and other real space functions such as energy density and the Laplacian of electron density at the so-called bond critical point (BCP) of the interatomic interaction could be comprehensively analyzed. The analysis of hydrogen bonding and non-covalent interactions was performed using Multiwfn (below Figure 5).

Figure 5. BCPs of the OSS network and OSSA network.

QTAIM analysis provides the direct evidence that the amide linkages in OSSA Network foster a superior hydrogen-bonding network compared to the ester linkages in OSS Network. For the characteristic N-H \cdots O hydrogen bond interaction in the amide-linked Network, the average electron density (ρ) at BCP is 0.025814 a.u., corresponding to a medium-strength hydrogen bond. In contrast, the strongest comparable interactions in the ester-linked Network exhibit a significantly lower average electron density of 0.022932 a.u. Using the empirical correlation of $E_{\text{HB}} = -223.08 \cdot \rho(\text{BCP}) + 0.7423$, the average hydrogen bond energy in OSSA Network is -21.95 kJ/mol, which is approximately 43% stronger than the strongest interactions found in OSS Network (-15.37 kJ/mol). This indicates that not only are there more hydrogen bonds in OSSA, but also the average strength of these hydrogen bonds is higher. In addition to strong N-H \cdots O hydrogen bonds, we also observe the weaker C-H \cdots O interactions. The slightly lower energy of these interactions in OSSA (-3.12 kJ/mol vs. -4.13 kJ/mol in OSS) is consistent with a more crowded molecular environment where the dominant strong amide-amide hydrogen bonds optimize the packing, leaving less configurations for the weaker C-H \cdots O interactions to form optimally. This overall shift towards a higher proportion of strongly cooperative hydrogen bonds within the dense OSSA network contributes to a more rigid and compact structure.

So, the computational analysis results unequivocally demonstrate that the simple change from an ester to an amide linkage in an otherwise identical octa-PEG architecture fundamentally alters the hydrogen bonding landscape. The amide groups in OSSA Network serve as potent sites for forming strong and cooperative N-H \cdots O hydrogen bonds, creating a robust and dense network of non-covalent crosslinks. This provides the atomic-level mechanism for the observed macroscopic superiority: the synergistic effect of covalent crosslinks and robustly dense hydrogen-bonding network severely restricts polymer chain mobility and creates a formidable barrier against the penetration of water and hydronium ions (e. g., H₃O⁺, enzyme). This directly explains the OSSA hydrogel's exceptionally low swelling ratio and its outstanding stability in the acidic gastric environment, where the ester-based network would be rapidly hydrolyzed. Therefore, this fundamental structure-property relationship, now clarified from the electronic to the macroscopic level, validates our molecular design strategy and reveals why OSSA is uniquely capable of achieving long-term sealing of large gastric defects, thereby highlighting its transformative potential.

3. The 3-cm porcine defect: a demonstration of transformative clinical potential

It is well-known that the gastric tissue, in fact, can quickly update the mucosa and accelerate the gastric healing with the assistance of bioadhesives in cases of gastric perforations defects with relatively small size (<1 cm), whereas larger gastric perforations are routinely challenged and rarely reported so far. In this work, we contend that the ability to seal a 3-cm gastric defect in a porcine model laparoscopically without additional sutures is not merely an incremental increase in size. To the best of our knowledge, this is the largest defect ever reported to be sealed by a hydrogel bioadhesive in a large animal model. It is a groundbreaking demonstration of a new clinical capability. It proves, for the first time, that a minimally invasive approach combined with laparoscopic-endoscopic technique can be effectively and safely applied to a severe gastric injury that currently mandates highly invasive open surgery. This

directly addresses a significant clinical unmet need and has the potential to alter surgical practice. Consequently, our work transcends the incremental by introducing a new class of stable bioadhesives specifically engineered to overcome the previously insurmountable challenge of long-term gastric sealing in large, critically sized defects. We have not only developed a new material but have also validated a groundbreaking therapeutic strategy for a condition that lacks advanced minimally invasive options. We believe this fulfills the journal's criterion for transformative research that can impact clinical practice and open new therapeutic avenues.

In summary, we have provided decisive new evidence to address the concern of novelty. First, direct benchmarking reveals that commercial tetra-PEG sealants and previously reported ester-linked octa-PEG system fail rapidly in gastric environments, whereas our amide-linked OSSA hydrogel maintains robust sealing, proving it is not an incremental improvement but a fundamental solution to a core instability problem. Second, QTAIM analysis quantitatively demonstrates that our octa-armed amide network fosters significantly stronger hydrogen bonding, elucidating the atomic-level mechanism for its superior performance. Third, the ability to seal a 3-cm porcine defect laparoscopically is the direct and unprecedented clinical validation of this transformative material design, moving beyond the capabilities of any previously reported system. Consequently, this work is not merely 'another bioadhesive' but a pivotal clinical advance that unlocks a new surgical capability—the minimally invasive management of large gastric defects, addressing a long-standing limitation in the field.

Reviewer #2 (Remarks to the Author):

The manuscript has improved substantially after the authors addressed the reviewers' comments. However, the schematic figures appear to have been generated using AI tools, and the source or method of generation has not been disclosed. The authors should provide clarification and include appropriate attribution in accordance with the journal's guidelines.

Reply: We sincerely thank the reviewer for their positive feedback on our revisions and for raising this important point regarding the schematic figures. We would like to clarify that all schematic illustrations in our manuscript were created by us directly using professional graphic software, and no AI-based image generation tools were used in the process. All elements in these schematic figures, including molecular structures, diagrams, and annotations, were manually designed and drawn to ensure their scientific accuracy and clarity. Specifically, these models were created using modeling software Blender and C4D, then rendered with Keyshot, and finally laid out and text edited in Photoshop.

In accordance with the journal's policy on image integrity, we hereby confirm that these are our original works. We have reviewed the journal's guidelines for figure preparation and confirm that our methods of creation and presentation are fully compliant. We have provided all original and editable source files to the editorial office for their verification.

Finally, we once again appreciate your thorough review of our manuscript with valuable suggestions to strengthen the impact and translational value, and hope this clarification can fully resolve the concerns and get your recognition. We recognize that this feedback is of great importance in further refining and enhancing our research outcomes.

A point-by-point rebuttal to the reviewer

Dear Reviewer,

Thanks very much for your kind advice and valuable comments in helping us improve our manuscript (Manuscript ID: NCOMMS-25-23781C). **All the modified words, sentences and paragraphs were labeled with red fonts.** A point-to-point response to highlight how we have addressed each of the comments is listed below.

Reviewer #1 (Remarks to the Author):

The authors have addressed some of the initial queries, but several critical issues remain unresolved, preventing me from recommending the article for publication. Regarding their claim that current hydrogel sealants, primarily based on hydrolytically labile bonds, cannot reliably endure stability in extremely acidic fluids and digestive enzymes, this assertion is contradicted by a substantial body of literatures. Numerous studies, such as *Science Translational Medicine* 12, 558 (2020): eaba8014; *Science advances* 7, no. 23 (2021): eabe8739; *Science translational medicine* 14, 630 (2022): eabh2857; *ACS nano* 17, 1 (2022): 111-126; *Advanced Functional Materials* 32, 29 (2022): 2202285; *Advanced Functional Materials* 34, 48 (2024): 2408479; *Chinese Journal of Chemistry*, 41, 23 (2023): 3339-3348; *ACS Nano* 17, 10 (2023): 9521-9528; *Science Translational Medicine* 17, 787 (2025): eadq1975; to cite only a few, demonstrate the successful use of hydrogel bioadhesives with long-term resistance to gastric fluids. A comprehensive review in *Theranostics*.16(2026):2221–2248 further supports this. Thus, the authors' claim that most hydrogels fail in such environments is unfounded, as viable alternatives are well-documented.

The authors also argue that adhesives capable of maintaining structural stability in the harsh gastric environment are either difficult to degrade or produce toxic degradation products. However, this generalization is problematic. Many of the cited successful systems neither face these challenges nor exhibit such limitations. The authors should re-examine these examples to avoid making sweeping and unsubstantiated claims about the degradability and toxicity of existing hydrogels.

While the authors highlight that their study involves the largest defect ever reported to be sealed by a hydrogel bioadhesive in a large animal model, this alone does not justify groundbreaking innovation. The increase in defect size does not inherently demonstrate why previous systems would fail at this scale or whether the limitation was merely due to the absence of such large models in prior studies. Without this justification, the novelty and

impact of their work remain unclear.

After a thorough review of the authors' response and the revised manuscript, I conclude that the study lacks sufficient strengths to meet the high standards of the journal. I regret to say that, the unresolved issues and unsubstantiated claims undermine the manuscript's potential for acceptance.

Reply: We are grateful to the reviewer for highlighting the need for greater precision in our comparison to the literatures. We acknowledge that our original claims regarding the limitations of existing hydrogels were overly broad and insufficiently nuanced. The key revision we have made is to shift the focus from a general critique of hydrogel stability to a detailed definition of the specific multi-property challenge our work addresses. We have clearly stated that although several hydrogel bioadhesives exhibit good gastric fluid resistance, the simultaneous integration of instant strong wet adhesion, low swelling, extreme acid/enzyme tolerance, long-term dynamic stability, durability, and suitable biodegradation into a single formulation for large-defect repair remains a significant hurdle. This integrated performance profile is essential for clinical translation but is rarely reported.

Moreover, we agree that our original words may be misinterpreted upon re-examination, and have removed all broad statements regarding the toxicity or non-degradability of the existing acid-resistant systems. Instead, we now frame the discussion around the trade-offs that many hydrogel systems face. Therefore, we have re-written the relevant sections to avoid sweeping generalizations, and used these cited literatures to construct a more accurate landscape, thereby determining the specific contribution of the proposed OSSA hydrogel to meet these various comprehensive requirements.

In addition, we demonstrated the repair of a full-thickness gastric defect (3 cm) in a porcine model, which represents one of the largest defects reported to be sealed by a hydrogel adhesive. We clarify that the significance of our work lies not merely in the defect size itself, but in demonstrating that our adhesive can meet the heightened demands at this clinically relevant scale that have rarely been tested or validated in previous studies. The repair of such clinically relevant large defects poses distinct challenges, including heightened exposure to gastric fluid flux, greater mechanical stress from peristalsis, and increased tension across the wound site. Success in this demanding scenario underscores our adhesive's exceptional adhesion strength, integrity, durability, and functional stability, moving beyond proof-of-concept in small models toward a scalable solution for serious clinical injuries.

Finally, we are deeply grateful to the reviewer for the continued engagement and constructive suggestions, especially regarding the contextualization of our research within

the existing literatures and guidance of potential translational values, have significantly enhanced the clarity, accuracy, and overall impact of our study. We believe that the comprehensive revisions described above, particularly the reframing of our claims, the more precise contextualization within the literatures, and the enhanced justification for the translational significance of large-defect repair, have fully addressed the remaining concerns. We respectfully submit that the revised manuscript presents a clear, accurate, and compelling case for its novelty and impact, and we hope the reviewer will find it suitable for publication in its current form.